# Identification and characterization of a ubiquitin E3 RING ligase of the *Chlamydia*-like bacterium *Simkania negevensis*

Eva-Maria Hörner[1], Vanessa Lachmayer[2], Thomas Hermanns[2], Adriana Moldovan[1], Kay Hofmann[2], Vera Kozjak-Pavlovic[1]*

1 Chair of Microbiology, Biocenter, Julius-Maximilian University Würzburg, Würzburg, Germany, 2 Institute for Genetics, University of Cologne, Cologne, Germany

* vera.kozjak@uni-wuerzburg.de

## Abstract

In the arms race between a pathogen and the host, the defense mechanisms of the host cell, including the ubiquitin system, are often counteracted by bacteria. *Simkania negevensis* (Sne), an obligate intracellular *Chlamydia*-like bacterium connected with respiratory diseases, possesses numerous deubiquitinases, but not much is known about its other ubiquitin-modifying enzymes. Sne infects a wide range of hosts, developing inside a tubular vacuole in close contact with the host endoplasmic reticulum (ER) and mitochondria. Our study describes an uncharacterized Sne ubiquitin E3 RING-ligase (SNE_A12920 or SneRING), which primarily generates K63- and K11-linked ubiquitin chains and preferentially interacts with UbcH5b and UBE2T E2 enzymes. SneRING is expressed upon infection of various human cell lines, as well as amoebae. We show that a portion of the expressed SneRING co-localizes with mitochondria and ER and that the SneRING interactome includes mitochondrial and ER proteins involved in organelle morphology and stress response. Our work offers an initial characterization of a bacterial RING ligase potentially involved in the host cell remodeling to accommodate the unique intracellular lifestyle of Sne.

## Author summary

Ubiquitination is a protein modification system that regulates protein degradation, localization, or interactions. As such, ubiquitination has many important functions in cell signalling, and its dysregulation can lead to cancer and neurodegenerative diseases. Bacteria that live and develop inside human or other eukaryotic cells, such as *Chlamydia,* often modulate the ubiquitination system to ensure their own survival. *Simkania negevensis* is a *Chlamydia*-like bacterium connected to respiratory diseases in humans. We have discovered a novel enzyme expressed by these bacteria that can ubiquitinate other proteins and thus potentially modify

**Data availability statement:** All relevant data are within the manuscript and its Supporting Information files.

**Funding:** This study was supported by the Deutsche Forschungsgemeinschaft (DFG, German Research Society) through a GRK2243/1 + 2 grant to VK-P, which included the salary of EMH. The funders had no role in study design, data collection and analysis, decision to publish, or preparation of the manuscript.

**Competing interests:** The authors have declared that no competing interests exist.

host cell processes that would otherwise hinder infection. In this work, we explore the function of this enzyme and determine its possible cellular localization, as well as some of the proteins it interacts with. Our study provides new insights into how bacterial pathogens adapt to and manipulate host cells using one of the major cell function regulatory systems.

## Introduction

*Simkania negevensis* (Sne) is an obligate intracellular *Chlamydia*-like bacterium, first described in 1993 as a cell culture contaminant of unknown origin [1]. Based on a sequence comparison of the ribosomal DNA (rDNA), it has been classified as a member of the family *Simkaniaceae* within the order *Chlamydiales* [2]. The bacterium survives and replicates in a variety of hosts. These range from different eukaryotic cells [3–5] to arthropods [6] and amoeba [7]. However, its natural host is unknown. Sne has been associated with different infections of the respiratory tract, including acute bronchiolitis in infants [8] and community-acquired pneumonia [9]. Furthermore, patients positive for Sne had a higher risk of transplanted lung rejection [10,11]. In addition, Sne has also been connected to Crohn's disease [12], and has been detected in the female genital tract [13]. However, recent studies dispute the link between Sne and respiratory tract infections [14]. While some of the closely related *Chlamydia*-species, such as *Chlamydia trachomatis*, the causative agent of sexually transmitted disease and trachoma [15], or *Chlamydia pneumoniae*, which causes acute respiratory diseases [16], are highly pathogenic, it remains unclear to what extent Sne can cause infections in immunocompetent individuals. Multiple studies of the role of Sne in respiratory diseases point to the conclusion that Sne exists rather as a commensal or, at best, as an opportunistic pathogen, occurring often in mixed infections [17–20].

Like other members of the *Chlamydiales*, Sne exhibits a biphasic developmental cycle, alternating between the replicative, metabolically active intracellular form called Reticulate Bodies (RB) and the infective environmental form called Elementary Bodies (EB) [1,21]. Inside the host cell, Sne replicates within a compartment called a "*Simkania*-containing vacuole" (SnCV). SnCV is a membranous, tubular system associated with mitochondria and forming extensive contact sites with the endoplasmic reticulum (ER) of the host cell [22,23]. Sne depends on the retrograde transport for nutrient acquisition and growth [24], with SnCV development and the number of RBs reaching a plateau on day 3 post-infection (p.i.) [21]. An increased release of Sne from infected cells was observed starting from day 4 p.i. [25], which is longer than the usual 2–3 days for *C. trachomatis* [26]. In addition, Sne-infected cells can be observed in culture for up to 15 days [21].

Sne is capable of prolonged intracellular growth while efficiently suppressing the host defense system, but little is known about how the bacterium accomplishes this. Sne can suppress the ER stress response [22], while infected cells show a strong resistance to apoptosis [27]. A recent report identified an unusually high number

of deubiquitinating enzymes (DUBs) in Sne belonging to classes of which some have never before been described in bacteria [28]. Considering that ubiquitination of cytosolic bacteria, as well as bacteria that live in specialized vacuoles, is a component of the cell-autonomous immunity combating intracellular pathogens [29], we aimed to further explore the ubiquitin-modifying enzymes of Sne.

Ubiquitination is a dynamic, versatile protein modification that regulates various cellular aspects in eukaryotes. The covalent attachment of ubiquitin to the targeted protein occurs through a cascade with three different enzymes: the E1 ubiquitin-activating enzyme, the E2 ubiquitin-conjugating enzyme, and the E3 ubiquitin-ligating enzyme that transfers ubiquitin to the substrate. During ubiquitination, a single ubiquitin molecule is attached to a lysine residue of a specific substrate through an isopeptide bond. The complexity of ubiquitination is further increased by the property of ubiquitin to become ubiquitinated at one or more of its seven lysine residues, creating ubiquitin chains. In addition, the N-terminus of ubiquitin can be modified, which results in eight possible linkage types [30].

The prototypical ubiquitin chains are linked by lysine (K) 48 or 63 of the ubiquitin moieties. While K48-linked ubiquitin chains target their substrate for proteasomal degradation, K63-linked ubiquitin chains regulate NF-κB transcription factor activation and immune response, as well as protein localization, DNA repair, and other important cellular processes. K11-, K6-, and M1-linked ubiquitin chains are considered atypical conjugates. K11-linked chains play a role during cell division, but are also associated with innate immune response against viruses [31]. K6-linked ubiquitin chains have been observed to accumulate upon depolarization of mitochondria and after UV radiation. Ubiquitin moieties linked via their free N-terminus, also referred to as linear or M1-linked ubiquitin chains, are involved in the regulation of immune response and inflammation through regulating the activation of the NF-κB transcription factor [32].

Based on their structure and mechanism of ubiquitin transfer to the substrate, E3 ubiquitin ligases are usually divided into three classes: RING (really interesting new gene), RBR (RING-between-RING), and HECT (homologous to the E6/AP carboxyl terminus) ligases. While RBR and HECT E3s form a catalytic intermediate with ubiquitin by binding the ubiquitin from the loaded E2 enzyme before conjugating it to the target protein, RING ligases, characterized by a RING domain, directly transfer ubiquitin from the E2 enzyme to the substrate [33]. With more than 600 RING ligases encoded by the human genome, this group is the largest [34,35].

Many pathogenic bacteria have acquired a sophisticated range of effectors, including ubiquitin E3 ligases, to exploit the host cell's ubiquitin system for their benefit. Examples include SopA, a HECT-like E3 ligase from *Salmonella* [36], LubX, a U-box-containing E3 ligase from *Legionella pneumophila* [37], as well as several novel E3 ligases of the IpaH family from *Shigella* and *Salmonella* [38,39]). By ubiquitinating the substrates, they modify the ubiquitin code to form a niche that allows them to survive and multiply within the host cell [40].

In this work, we looked for putative E3 ubiquitin ligases of Sne. We focused on a RING ligase SNE_A12920 (SneR-ING), a 25.4 kDa protein containing an unusual RING finger when compared to other bacterial RING ligases. We purified the enzyme and analyzed its activity and the type of ubiquitin chains it generates. We show that SneRING is expressed in different human cell lines as well as in amoeba during infection. After expression in human cells, although the majority of the SneRING is found in the cytosol, we show that a portion of the protein associates with the ER and mitochondria. Possible SneRING interaction partners we identified include mitochondrial and ER proteins playing a role in organelle morphology, protein degradation, and stress response, indicating the potential mode of action of this bacterial enzyme. Taken together, our data provide first insights into the previously unknown world of ubiquitin ligases in *Simkania negevensis*.

## Results

### Identification of RING ligases in *Simkania negevensis*

To identify ubiquitin ligase candidates in the sequenced genome of Sne [41], the Sne subset of UniProt [42] was searched with generalized profiles [43] representing classical RING finger domains and several RING finger variants. When using a

profile derived from a multiple alignment of established RING domains, only a single Sne ORF, SNE_A12920 (SneRING), reached a significant score of p < 0.01. To validate this finding, BLAST and profile searches were used to find other bacterial proteins related to SneRING. A small family was identified, of which representative members include Sim-KFB93 and Sim-QNJ27 from bacteria belonging to the Simkaniaceae family, and Chl-HKST.2, belonging to an unclassified Chlamydiia (Fig 1A). An Alphafold [44] model of SneRING shows a two-domain architecture with a RING-like fold forming the first domain (residues 14–72), followed by a linker region and a C-terminal α/β-fold domain without informative sequence or structural similarities (Fig 1B). Like canonical RING fingers, the RING-like domain of SneRING is predicted to coordinate two $Zn^{2+}$ ions, albeit using somewhat atypical ligand residues. In the model, the first $Zn^{2+}$ ion is coordinated by Cys-17, Cys-20, His-40, and Cys43, while the second $Zn^{2+}$ ion is contacted by Asn-32, Cys-35, Cys-54, and Cys57.

A DALI [45] search using the RING-like region of SneRING identified as significant top-hits the established RING fingers of TRIM37 (pdb:3LRQ, Z-score 9.1) and RING2 (pdb:6WI7, Z-score 8.9). RING2 possesses a typical RING finger, where $Zn_1$ is coordinated by two CxxC dyads and $Zn_2$ is bound by a CxH and a CxxC dyad. In SneRING, both coordination spheres are slightly altered: His-40 replaces Cys as the 3$^{rd}$ ligand of $Zn_1$, while the first two ligands of $Zn_2$ are NxxC

**A**

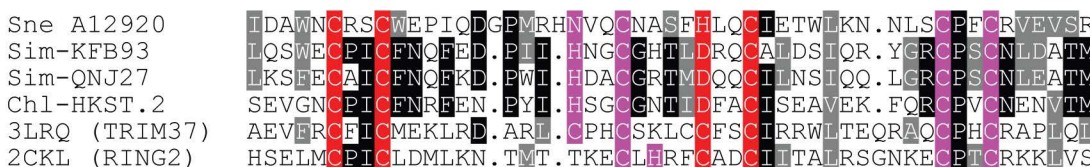

```
Sne A12920   IDAWNCRSCWEPIQDGPMRHNVQCNASFHLQCIETWLKN.NLSCPFCRVEVSR
Sim-KFB93    LQSWECPICFNQFED.PII.HNGCGHTLDRQCALDSIQR.YGRCPSCNLDATM
Sim-QNJ27    LKSFECAICFNQFKD.PWI.HDACGRTMDQQCILNSIQQ.LGRCPSCNLEATM
Chl-HKST.2   SEVGNCPICFNRFEN.PYI.HSGCGNTIDFACISEAVEK.FQRCPVCNENVTM
3LRQ (TRIM37) AEVFRCFICMEKLRD.ARL.CPHCSKLCCFSCIRRWLTEQRAQCPHCRAPLQL
2CKL (RING2)  HSELMCPICLDMLKN.TMT.TKECLHRFCADCIITALRSGNKECPTCRKKLVS
```

**B**                                                          **C**

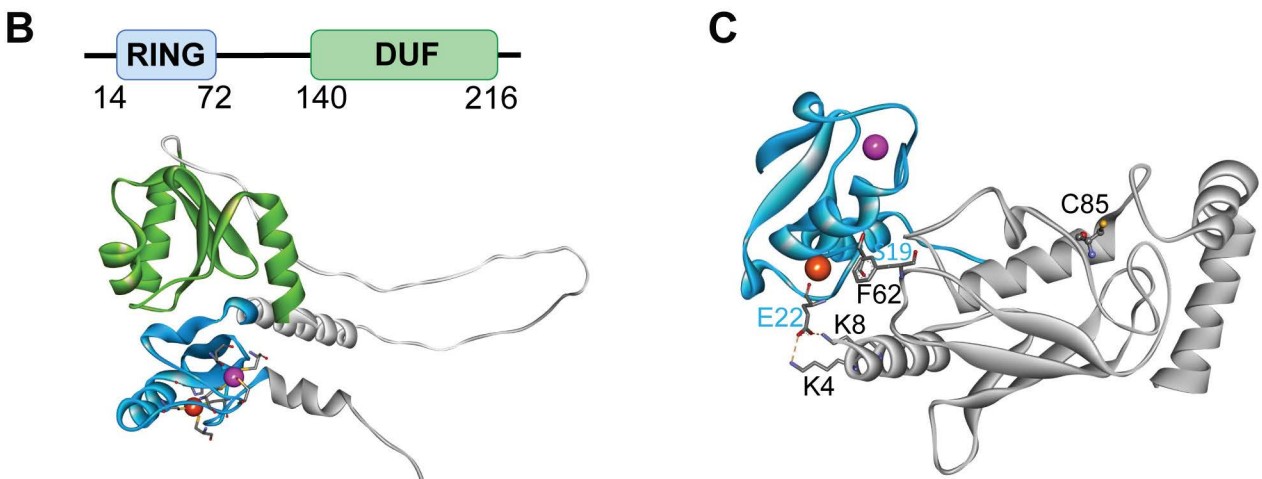

**Fig 1. SneRING is a predicted RING domain ligase. (A)** Multiple alignment of the RING domains of SneRING (SNE_A12920), some bacterial relatives (Sim-KFB93: Simkaniaceae bacterium RefSeq:QVL56959; Sim-QNJ27: Simkaniaceae bacterium RefSeq:MDJ0651792; Chl-HKST: Chlamydiia bacterium Uniprot: A0A960X5Q0), and sequences of the two best DALI hits (pdb:3LRQ (TRIM37 from *Homo sapiens*) and pdb:2CKL (RING2 from *Mus musculus*)). Residues invariant or conserved in at least 50% of the sequences are shown on black and grey background, respectively. Residues involved in the coordination of $Zn_1$ and $Zn_2$ are highlighted in red and magenta, respectively. **(B)** Alphafold model of SneRING shows the N-terminal RING-like domain (blue) and a C-terminal domain of unknown function (DUF, green). The $Zn_1$ and $Zn_2$ ions coordinated by the RING-like domain are shown in red and magenta, respectively. **(C)** Alphafold model of the RING domain of SneRING (blue) in contact with UbcH5b (grey). The $Zn_1$ and $Zn_2$ ions coordinated by the RING-like domain are shown in red and magenta, respectively. The predicted contact residues Glu-22 (E22) and Ser-19 (S19) and their interaction partners are labeled.

rather than the canonical CxH (Fig 1A). Functionally, these changes might be inconsequential, as the Alphafold model of the SNE_A12920 RING domain in contact with UbcH5 predicts SneRING to form a complex with E2 enzymes resembling that of other RING ligases (Fig 1C). As with other RING fingers, the most important contacts are made by the region surrounding the first Zn-binding dyad. While most other RING domains contact E2s by a conserved hydrophobic residue before the second Zn-binding cysteine, SneRING carries the atypical Ser-19 at this position, which does not form a specific E2 interaction in the model. Instead, Glu-22 forms short-distance salt bridges to Lys-4 and Lys-8 of UbcH5 (Fig 1C). We reasoned that a mutation of Glu-22 combined with a Ser-19 mutation might abrogate E2-binding. Indeed, when subjecting an S19R_E22R variant of SneRING to Alphafold modeling with UbcH5, no productive complex was formed.

Besides the significantly-scoring SneRING, three further RING-like domains could be identified bioinformatically. A closer inspection of the borderline-significant matches identified the uncharacterized ORF SNE_A08700 and its close relative (and genomic neighbor) SNE_A08690 to have a conspicuous arrangement of cysteine and histidine residues. The assumed RING-relationship of these two proteins is supported by the conservation pattern of their sequence family and by Alphafold modeling: SNE_A08700 is predicted to use eight canonical ligands for coordinating two $Zn^{2+}$ ions (S1A Fig); the best DALI match to an established RING-finger was RNF125 (pdb:5DKA, Z-score 5.6). Sequence conservation between RNF125 and the SNE_A08700 family shows a big insertion after the 3rd Zn ligand (S1B Fig), which explains the poor score in the initial profile search. The closely related protein SNE_A08690 has lost three ligands for the first $Zn^{2+}$ ion and probably coordinates only a single ion. A 4th RING candidate (SNE_A19470) was picked up in profile searches using the SP-RING family, a subgroup of RING-like domains usually involved in SUMO conjugation [46]. However, a closer inspection suggested that SNE_A19470 does not contain an SP-RING domain but rather a (degenerate) conventional RING that has lost its ligands for $Zn_1$ coordination (S1C Fig). The best DALI matches were TRIM69 (pdb:6YXE, Z-score 8.5) and TRIM2 (pdb:8A38, Z-score 8.3), which both contain a structurally similar RING domain, but with a complete set of eight ligand residues (S1D Fig).

Considering that SneRING was the only protein with a RING-like domain that has been identified in a previously published proteomics analysis of the ER/SnCV membrane of infected HeLa229 cells on day 3 p.i. [24], we focused on further characterization of this enzyme.

### SneRING is an active E3 RING ligase with an E2 enzyme-interacting region within its predicted RING domain

We next tested the activity of the SneRING using an *in vitro* autoubiquitination assay. For this, we purified SneRING and incubated it with a ubiquitin-activating (E1) and a ubiquitin-conjugating enzyme (E2), as well as free ubiquitin and ATP to allow the enzyme cascade to covalently attach ubiquitin to SneRING. Already after 30 min of incubation, we could detect a strong ubiquitin signal that increased over time, while in the negative control without SneRING, only monoubiquitin could be detected (Fig 2A). The resulting signal represented an autoubiquitinated SneRING because a similar pattern could be observed when the reaction was analyzed using SneRING antibodies (S2A Fig). This shows that the SneRING is an active ubiquitin ligase.

Since Alphafold modeling indicated the importance of Ser-19 and Glu-22 for the interaction with an E2 enzyme, we replaced these two amino acids with arginine to generate a catalytically inactive form and performed an *in vitro* autoubiquitination assay with the purified mutated enzyme for 5 h. While a strong ubiquitin signal could be detected in the presence of the wild-type SneRING, no ubiquitin chains were formed in the reaction containing SneRING S19R_E22R (Fig 2B). This confirms that the N-terminal domain of the SneRING interacts with the E2 ubiquitin-conjugating enzyme and is necessary for its function.

To test whether SneRING can generate ubiquitin chains when combined with other E2 enzymes besides UbcH5b, we performed an *in vitro* autoubiquitination assay for 5 h using selected ubiquitin-conjugating enzymes. Only UbcH5b- and, to a lesser extent, UBE2T-containing reactions showed the presence of the polyubiquitin signals, while no ubiquitin chain assembly occurred with E2 enzymes UbcH7, UBE2A, UBE2B, and UBE2W (Fig 2C). The time course analysis of the

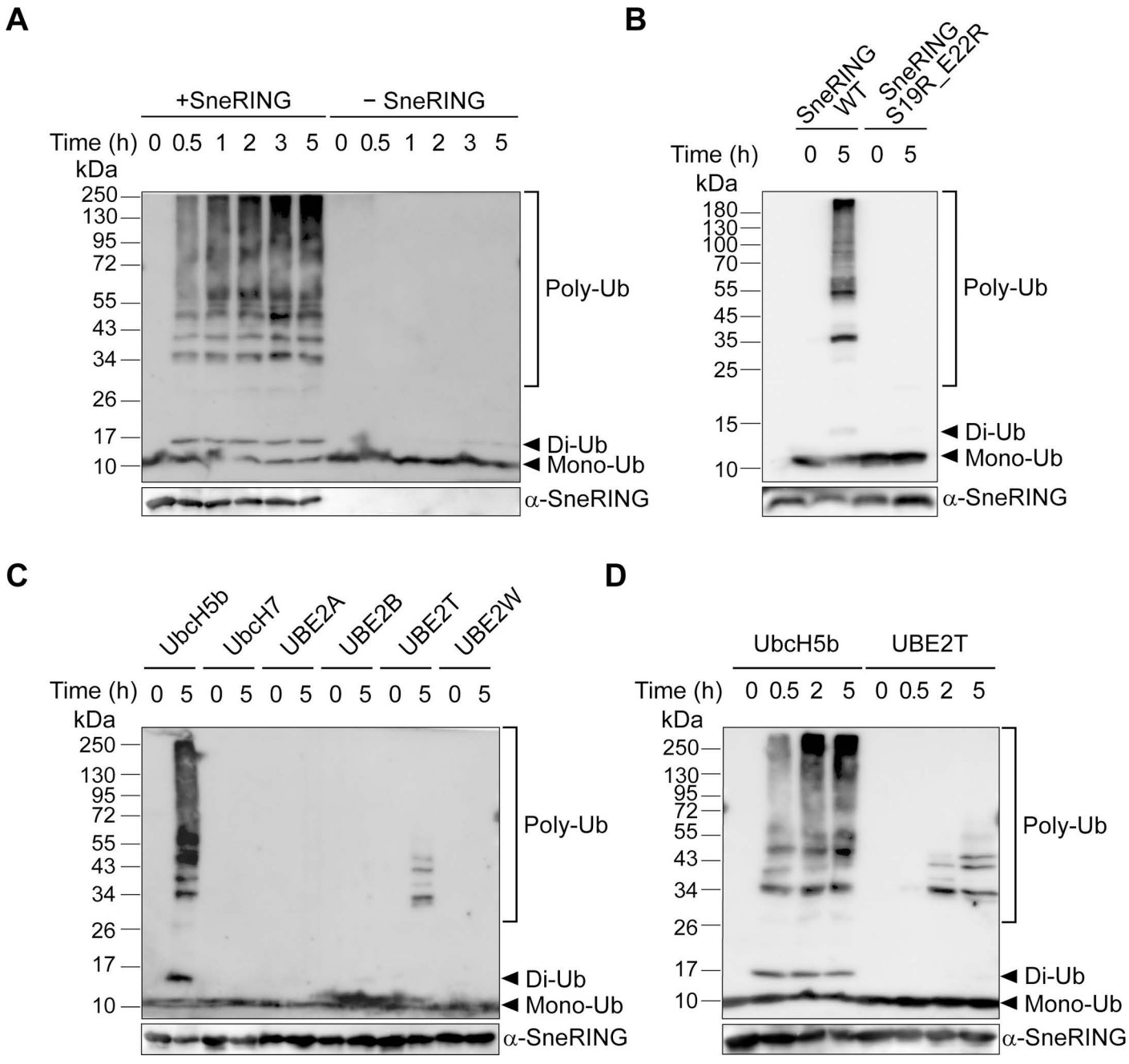

**Fig 2. SneRING shows ubiquitination activity in an *in vitro* assay.** (A) Purified recombinant SneRING was mixed with recombinant E1 ubiquitin-activating enzyme, E2 ubiquitin-conjugating enzyme (UbcH5b), ubiquitin, and ATP for the indicated periods. A reaction without the ligase served as a control. Samples were analyzed by SDS-PAGE and western blot, using an antibody against ubiquitin. **(B)** *In vitro* autoubiquitination reaction of wild-type SneRING and SneRING-ligase with mutated E2 enzyme interacting region (SneRING S19R_E22R) was performed for 5 h as in A and analyzed by SDS-PAGE and western blot, using anti-ubiquitin antibodies. **(C)** *In vitro* autoubiquitination reaction of wildtype SneRING was performed as in A in combination with indicated E2 enzymes for 5 h, followed by SDS-PAGE and western blot analysis using ubiquitin antibody. **(D)** SneRING was incubated with E2 enzyme UbcH5b or UBE2T in an *in vitro* autoubiquitination assay as described in **A**. Samples were taken at indicated time points and tested by immunoblot using the primary antibody against ubiquitin. Poly-Ub, polyubiquitin; Di-Ub, diubiquitin; Mono-Ub, monoubiquitin.

autoubiquitination reaction in combination with UbcH5b showed strong ubiquitination after 30 min of incubation, increasing over time (Fig 2A and 2D). In contrast, in the presence of UBE2T, nearly no reaction was observed after 30 min, and some polyubiquitin signals were detectable only 2 h and 5 h later (Fig 2D). In conclusion, among the tested E2 enzymes, SneR-ING appears to be most active in combination with UbcH5b, while a moderate reaction also occurs in the presence of the ubiquitin-conjugating enzyme UBE2T.

## Ubiquitin chains generated by SneRING are cleaved by deubiquitinating enzymes specific for the K11/K63 linkage type

To characterize the type of ubiquitin chains formed by SneRING, we performed linkage-specific chain-cleaving assays (UbiCRest assays) [47]. Ubiquitin chains generated by SneRING after *in vitro* autoubiquitination for 5 h were incubated with a panel of different DUBs that either process any chain type, such as ubiquitin-specific peptidase 21 (USP21) [48], or have specificity against certain ubiquitin chain linkages. When incubating the reaction with the USP21, no remaining polyubiquitin signal could be detected. Treatment with the K11-specific DUB Cezanne [49] or with the K63-specific DUB AMSH (associated molecule with an Src homology 3 domain of signal transducing adaptor molecule, STAM) [50] reduced the polyubiquitin signal. In contrast, no cleavage was observed upon incubation with the DUB SneOTU that specifically cleaves linear M1 ubiquitin chains or SneVTD that shows cleavage specificity against K6-linked ubiquitin chains [28]. Furthermore, also the K48-specific DUB OTUB1 [51] did not visibly reduce the ubiquitin signal (Fig 3A).

To test whether the ligase only generates these two linkage types or if other ubiquitin chain types are produced at a lower level, a di-UbiCRest assay was performed. For this, an *in vitro* SneRING autoubiquitination reaction was incubated with the K11-specific DUB Cezanne, with the K63-specific DUB AMSH, or with both enzymes simultaneously. As already observed, Cezanne and AMSH led to a reduction of the ubiquitin signal, demonstrating the presence of K11- and K63-linked ubiquitin chains. However, the combined activity of both DUBs did not completely degrade ubiquitin chains generated by the SneRING (Fig 3B), indicating that other linkage types are formed by the SneRING in addition to K11- and K63-linked ubiquitin chains. Mass spectrometry of the *in vitro* SneRING autoubiquitination reaction in the presence of UbcH5b (UBE2D2) mainly detected K11- and K63-linked GlyGly-modified peptides, with about 5-fold lower signal detected for K6- and K48-modified peptides, along with small amounts of K33-modified peptides (Fig 3C). Similar results with overall weaker signal strength were obtained for the reaction containing the E2 enzyme UBE2T (Fig 3D). In conclusion, SneRING forms *in vitro* mostly K11- and K63-, in addition to K6- and K48-linked ubiquitin chains.

To further validate this finding, *in vitro* autoubiquitination assays performed with UbcH5b were tested by immunoblot analysis for the presence of specific ubiquitin chains. We could confirm the presence of K63- (S2B Fig), K48- (S2C Fig), K11- (S2D Fig) and K6-linked ubiquitin chains (S2E Fig). However, when we used an antibody against M1-linkages, which were not detected by UbiCRest assay and mass spectrometry, we only saw a signal in the positive control, but not in the SneRING reaction, confirming the previous results (S2F Fig).

## SneRING is expressed during the infection of human cells and its possible natural host, *Acanthamoeba castellanii*

We next analyzed the expression pattern of SneRING during infection. Since Sne can infect and grow in a variety of cells [52], as well as survive and multiply within *Acanthamoeba castellanii*, we measured the SneRING mRNA levels during the infection cycle of various cell lines and amoeba by RT-qPCR analysis and normalized it against the levels of the bacterial 5S RNA. In comparison, we measured the mRNA levels of a highly expressed gene, Sne chaperone SnGroEL. SneR-ING was expressed in all host cells we tested, though to a lesser extent than SnGroEL (S3 Fig). Expression intensities of SneRING in different human cell lines were comparable. In HeLa229 cells, we observed somewhat higher SneRING expression on day 2 p.i. compared to day 4 p.i. (S3A and S3B Fig).

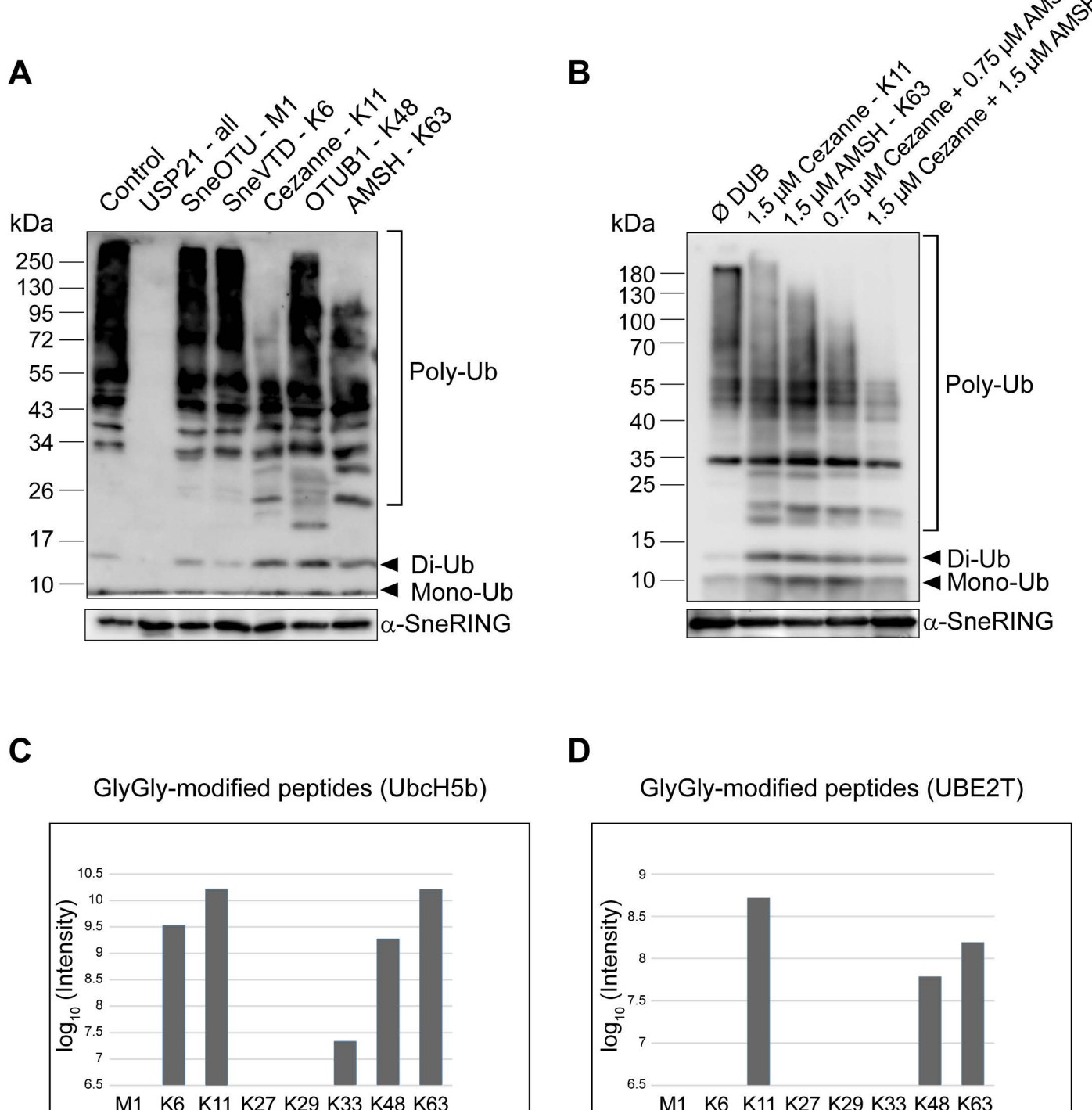

**Fig 3. SneRING generates K6-, K11-, K48-, and K63-linked ubiquitin chains that can be digested by specific DUBs. (A)** An *in vitro* autoubiquitination reaction, as in Fig 2A, was incubated for 5 h, after which a UbiCRest assay was performed using the indicated chain type-specific DUBs for 1 h at 37 °C. All DUBs were present at a final concentration of 1.5 µM, except OTUB1, which was present at 3 µM. Ubiquitin chain cleavage was analyzed by SDS-PAGE and western blot using a primary antibody against ubiquitin and SneRING as a control. **(B)** An *in vitro* autoubiquitination reaction, as in A, was incubated with Cezanne (K11-linked ubiquitin chain-specific DUB) and AMSH (K63-linked ubiquitin chain-specific DUB). Degradation of ubiquitin chains was detected by SDS-PAGE and western blot using anti-ubiquitin antibody. Poly-Ub, polyubiquitin; Di-Ub, diubiquitin; Mono-Ub, monoubiquitin.

**(C, D)** A 5 h *in vitro* autoubiquitination reaction of SneRING with UbcH5b (C) or UBE2T (D) was analyzed by mass spectrometry. The graphs show $\log_{10}$ intensities of GlyGly-modified residues for the different linkage types, as determined by MaxQuant.

Sne can infect and grow in the monocyte/macrophage cell line U937 [3], as well as in the THP-1 cell line that mimics the macrophage-like state after differentiation by PMA [53], in contrast to *C. trachomatis* [54]. We analyzed and compared the expression of SneRING in the THP-1 cell line and primary human M2-like macrophages. Based on the observation that infectious Sne particles can be detected on day 6 p.i. in U937 cells [3], we included this time point in our experimental setup. RT-qPCR analysis showed slightly higher expression of SneRING in THP-1 cells than in M2-like macrophages. There were no notable differences in the mRNA levels between different days p.i. (S3C and S3D Fig).

By taking up a high number of bacteria, many free-living protozoa mimic professional phagocytes [55]. Sne can survive and grow within amoebae [7]. Since Sne establishes a long-term infection in amoebae, we measured the mRNA level of SneRING only on day 4 p.i. *A. castellanii* culture was cultivated at 30 °C instead of 37 °C, which we used for the cultivation of human cell lines (or 35 °C for Sne-infected). mRNA of SneRING was detectable, but the levels varied strongly between the samples, both for SneRING and SnGroEL (S3E Fig).

### SneRING localizes to the host cell cytoplasm, mitochondria, and ER after overexpression

We next studied the localization of SneRING in the host cell. Since we were unsuccessful in producing an antibody against SneRING that would recognize the native protein in immunofluorescence studies, we assayed the FLAG-tagged protein expressed from a plasmid in U2OS cells. SneRING was successfully expressed and exhibited cytosolic distribution as observed by immunofluorescence microscopy (Fig 4A). The FLAG-tagged protein could also be identified by western blot after transfection (Fig 4B). Upon subcellular fractionation and separation of the cells into the crude mitochondrial (containing mitochondria and ER), the light membrane (containing ER, Golgi, endosomes, and plasma membrane), and the cytosolic fraction, SneRING signal was distributed between all fractions (Fig 4C). In comparison, Mic60, a mitochondrial inner membrane protein, localized almost exclusively to the mitochondrial fraction, while calnexin, the integral protein of the ER, was detected in both the light membrane and the crude mitochondrial fraction. GAPDH and tubulin were associated with every fraction but mainly localized in the cytosol, somewhat resembling SneRING distribution (Fig 4C).

Immunofluorescence microscopy of HeLa229 cells with GFP-labeled mitochondria (S4A Fig) or dsRED-labeled ER (S4B Fig) that were transfected with FLAG-tagged SneRING gave a similar picture, with some overlap of the signal, indicating possible association of SneRING with these organelles. This did not significantly change upon infection with Sne (S4A and S4B Fig, lower panels). Notably, Sne-infected cells show fragmentation of the mitochondria, which is absent in cells transfected only with SneRING. In both cell lines, the FLAG-tagged protein is pushed to the cell periphery with the cytosol and is not identified inside the SnCV when the cells are infected (S4 Fig).

The observed association of SneRING with mitochondria could be caused by the interaction with the mitochondrial surface or result from the insufficient purity of the mitochondrial fraction, which contains ER and contaminations from other cellular compartments. To analyze this further, we isolated mitochondria from the FLAG-SneRING-transfected Hek293T cells and subjected them to swelling in a hypotonic buffer in the absence or the presence of Protease K (PK). The accessibility of the proteins to PK demonstrates their submitochondrial localization. As a positive control, mitochondria were completely solubilized by 1% Triton X-100 and treated with PK (Fig 4D). Surprisingly, FLAG-tagged SneRING could be detected even after PK treatment of intact mitochondria, indicating that the portion of the protein is not localized on the mitochondrial surface. In contrast, Tom70, a mitochondrial outer membrane protein, is degraded under similar conditions, comparable to the ER protein calnexin. Opening of the outer membrane by swelling leads to a partial release of an intermembrane-space protein coproporphyrinogen oxidase (CPOX), but the inner membrane protein Mic60, the matrix protein Tim44, and SneRING remain undisturbed. PK treatment of the mitochondria with the ruptured outer membrane

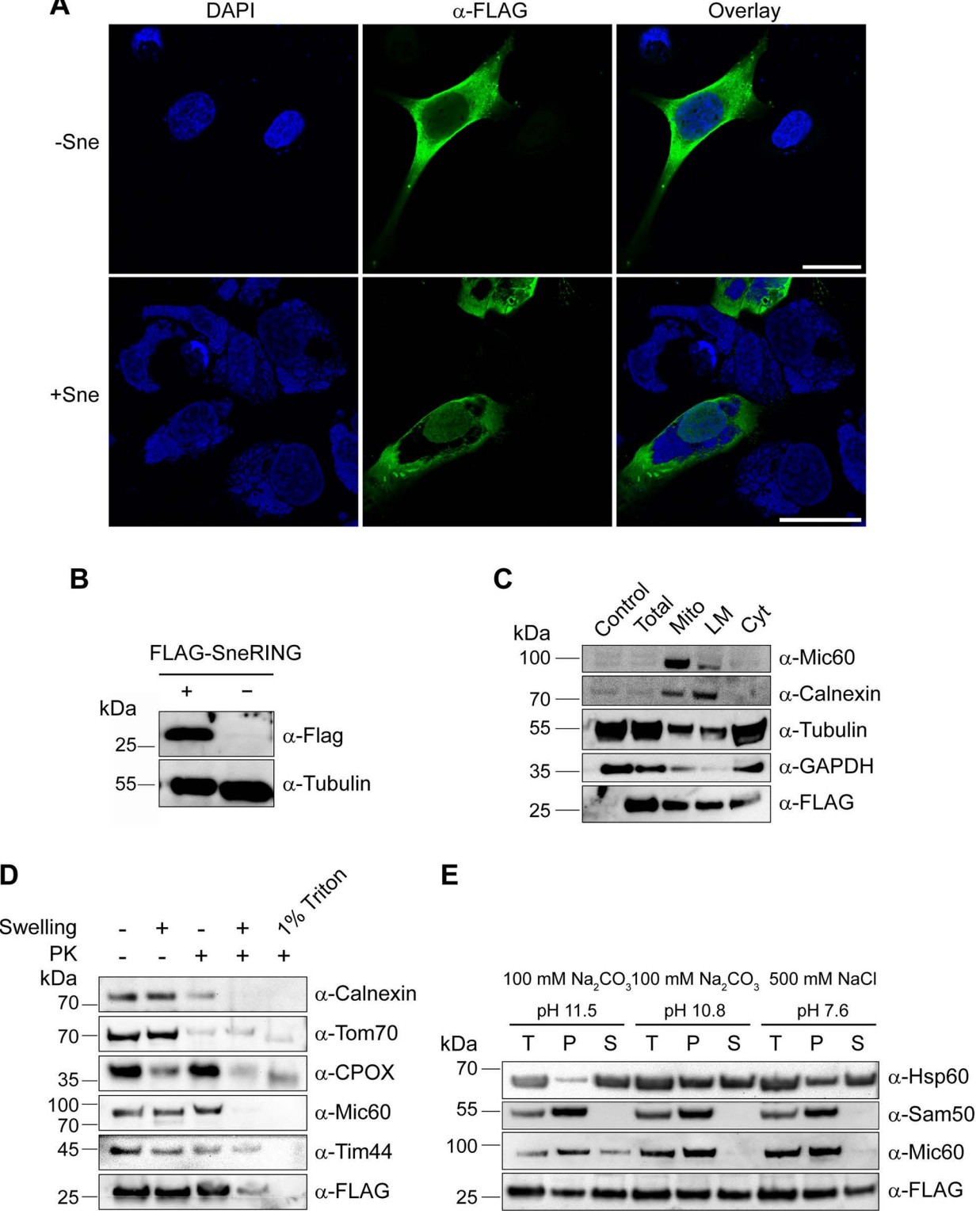

**Fig 4. SneRING localizes to the host cell cytosolic, light membranes, and inner mitochondrial membrane after overexpression. (A)** U2OS cells were transfected with a FLAG-tagged version of the SneRING using the transfection reagent PEI MAX. 72 h post-transfection and 2 days p.i., cells were fixed and stained using DAPI (blue channel), and a primary antibody against FLAG followed by the fluorophore-coupled secondary antibody (green

channel). Images were taken using laser scanning confocal microscopy. The scale bar represents 30 μm. **(B)** U2OS cells were transfected as in A and, 36 h post-transfection, analyzed by SDS-PAGE and western blot using antibodies against the FLAG-tag and tubulin. **(C)** U2OS cells were transfected with FLAG-SneRING as in A, and 48 h post-transfection, the cells were separated by differential centrifugation into a crude mitochondrial (Mito), light membrane (LM), and cytosolic (Cyt) fraction. Total cell lysate of non-transfected cells (Control) and transfected cells (Total) served as controls. The samples were analyzed by SDS-PAGE and western blot using antibodies against Mic60, Calnexin, Tubulin, GAPDH, and FLAG. **(D)** Hek293T cells were transfected with FLAG-SneRING using the calcium-phosphate method. After 48 h of expression, mitochondria were isolated and then incubated in isotonic buffer (-Swelling) or in hypotonic buffer (+Swelling) to rupture the outer mitochondrial membrane, combined with the treatment using 50 μg/mL of protease K (PK). For complete solubilization, mitochondria were treated with 1% Triton X-100 (1% Triton) and PK and precipitated using 72% trichloroacetic acid. Samples were analyzed by SDS-PAGE and western blot, using antibodies against Calnexin, Tom70, CPOX, Mic60, Tim44, and FLAG-tag. **(E)** Mitochondria as in D were either sonicated in a high-salt buffer (500 mM NaCl pH 7.6) or subjected to carbonate extraction (100 mM $Na_2CO_3$ pH 10.8 or 100 mM $Na_2CO_3$ pH 11.5). Pellet (P) and supernatant (S) fractions were separated by ultracentrifugation, while untreated total samples (T) served as controls. Samples were analyzed by SDS-PAGE and western blot using antibodies against Hsp60, Sam50, Mic60, and FLAG. GAPDH, glyceraldehyde-3-phosphate dehydrogenase; Tom70, translocase of the outer mitochondrial membrane 70; CPOX, coproporphyrinogen oxidase; Mic60, mitochondrial contact site and cristae organizing system 60; Tim44, translocase of the inner mitochondrial membrane 44; Sam50, sorting and assembly machinery 50.

leads to a degradation of CPOX as well as Mic60, which is exposed to the intermembrane space, whereas Tim44 is degraded completely only after complete solubilization of mitochondria by a detergent. SneRING behaves in this respect like Mic60 since most of the protein is degraded upon PK treatment of swollen mitochondria. A small portion of the protein, however, remains undigested, indicating possible protection by the inner mitochondrial membrane (Fig 4D).

We performed carbonate extraction of isolated mitochondria using 100 mM $Na_2CO_3$, pH 11.5 or pH 10.8 to assess membrane integration of SneRING. In addition, the membrane association of the protein was tested by sonicating mitochondria in a high salt buffer. Hsp60, a soluble mitochondrial matrix protein, was found in both pellet and supernatant fractions after milder carbonate extraction and extraction in the high-salt buffer, but was fully extracted at pH 11.5. In contrast, Sam50, an integral protein of the outer mitochondrial membrane, was only detected in the pellet after treatment in all three conditions. Mic60, the transmembrane protein of the inner mitochondrial membrane, was partially extracted from membranes only after carbonate extraction at a higher pH of 11.5. FLAG-tagged SneRING was detected in both the pellet and the supernatant after incubation in all three conditions, suggesting that the protein is mostly membrane-associated. Its distribution resembles that of Mic60, which possesses a membrane anchor, so it would appear that a small portion of SneRING could be a membrane-integrated protein (Fig 4E), though this could also be an artifact resulting from overexpression. We conclude that a portion of SneRING localizes to the host cell mitochondria and ER after overexpression. The mitochondrial protein is most likely associated with the inner mitochondrial membrane.

### Putative SneRING-ligase interacting partners include mitochondrial and ER proteins involved in the regulation of organelle morphology, protein degradation, and stress response

To determine possible host cell targets of SneRING, we expressed the FLAG-tagged protein in U2OS cells with and without Sne infection and performed a pull-down using the FLAG antibody, followed by mass spectrometry analysis. We compared infected samples with and without ectopically expressed FLAG-tagged SneRING to determine significantly enriched proteins (Fig 5A). Highly enriched proteins (log2 fold change > 2) included mitochondrial proteins involved in organelle morphology (prohibitins (PHB1 and 2), Mic60/IMMT), oxidative phosphorylation (OXPHOS) (subunits of ATPase and respiratory chain components), as well as metabolite transport (VDAC1 and 2). However, several ER proteins were also identified, and these included ER-microtubule anchoring protein CKAP4, BCAP31, a protein involved in ER-associated degradation (ERAD), and acting as an ER stress sensor that mediates communication with mitochondria [56], and Erlin1 and 2, which are prohibitin-domain containing proteins involved in ERAD [57] (Fig 5A and S1 Table).

When comparing the cellular SneRING-associated proteins of Sne-infected and non-infected cells, a very similar set of proteins was obtained. The top fourteen cellular infection-specific SneRING interactors were also found on the previous list. We again found mitochondrial and ER proteins among the hits. Interestingly, GOLIM4, a protein localized to the

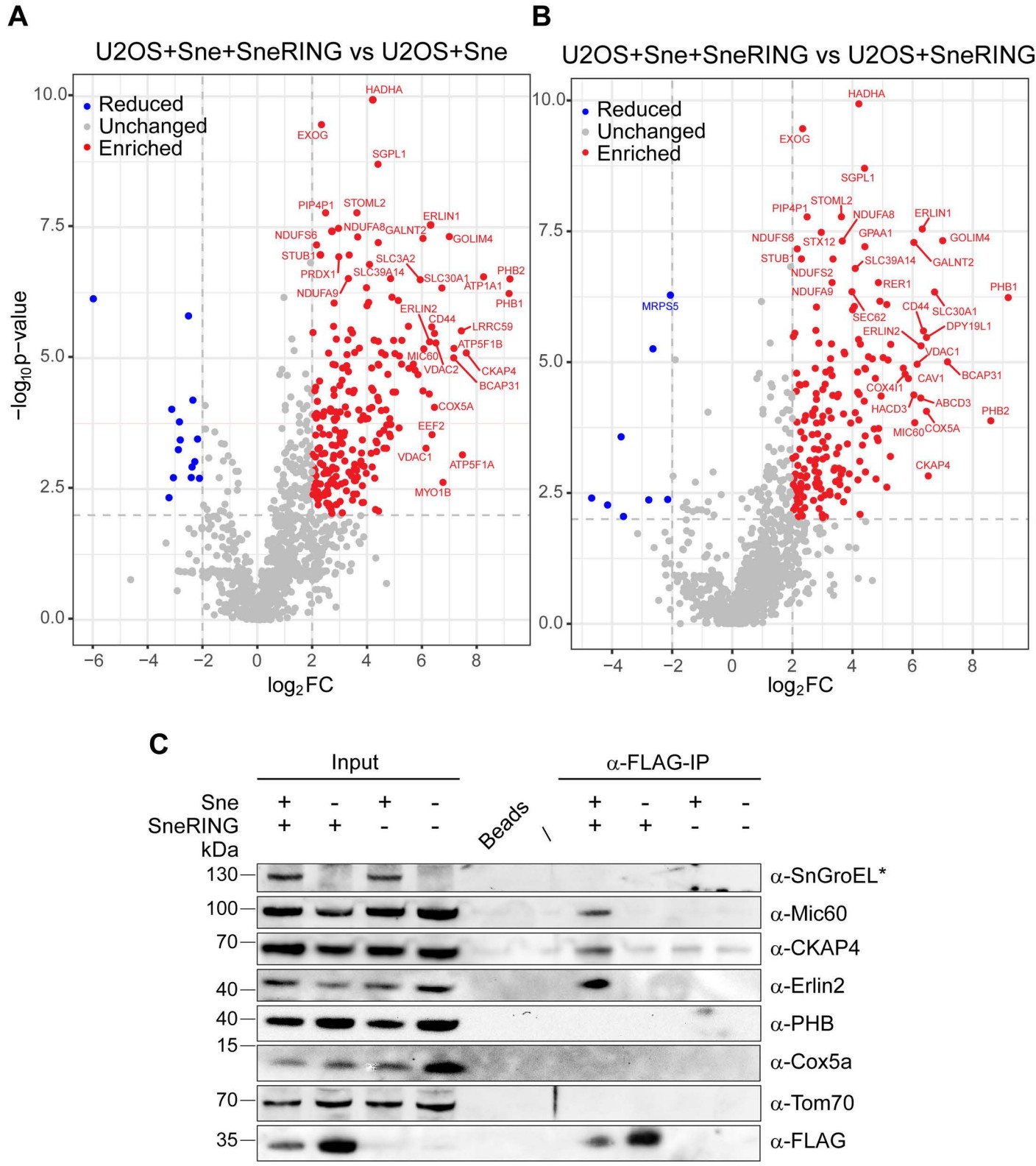

**Fig 5. SneRING interacts with mitochondrial and ER proteins. (A, B)** A FLAG-tagged version of the SneRING was overexpressed in U2OS cells using PEI MAX transfection. 24 h later, cells were infected with Sne at an MOI of 1. Non-transfected cells served as a control. 48 h p.i., samples were

collected and immunoprecipitation was performed using FLAG-magnetic beads, followed by mass spectrometry analysis. The graphs show identified proteins, with significance (-$\log_{10}$p-value calculated by two-tailed T-test, n = 3) plotted against the $\log_2$ fold change ($\log_2$FC) of transfected/infected U2OS cells (U2OS+Sne + SneRING) relative to non-transfected/infected controls (U2OS+Sne) **(A)** or of transfected/infected U2OS cells (U2OS+Sne + SneRING) relative to transfected/non-infected controls (U2OS+SneRING) **(B)**. Enriched proteins are labeled in red, reduced proteins are shown in blue, and grey represents unchanged proteins. Only host cell proteins are shown. **(C)** Samples were prepared, and immunoprecipitation was performed as in A. Input and elution fractions were analyzed by SDS-PAGE and western blot using antibodies against Sne heat shock protein SnGroEL (star indicates that the signal represents a cross-reactive band observed at 130 kDa), Mic60, CKAP4, Erlin2, PHB, Cox5a, Tom70, and FLAG. Mic60, mitochondrial contact site and cristae organizing system 60; CKAP4, cytoskeleton-associated protein 4; Erlin2, ER Lipid Raft Associated 2; PHB, Prohibitin; Cox5a, cytochrome *c* oxidase subunit 5A; Tom70, translocase of the outer mitochondrial membrane 70.

Golgi apparatus and playing a role in endosome to Golgi protein transport [58], was also found as one of the most highly enriched proteins in both lists (Fig 5B and S1 Table).

Western blot analysis of the pull-down samples confirmed the presence of Mic60/IMMT, CKAP4, and Erlin2, but PHB or Cox5a could not be detected despite being identified by mass spectrometry. Interestingly, Mic60/IMMT, CKAP4, and Erlin2 were primarily present in the pull-downs of Sne-infected samples expressing FLAG-tagged SneRING, indicating that active infection is necessary for the interactions with SneRING to take place (Fig 5C).

To further confirm that SneRING is capable of ubiquitinating some of the identified putative interactors, we performed an *in vitro* ubiquitination assay of CKAP4 in the presence of SneRING. In a control reaction, we used GST as an SneRING substrate. Whereas SneRING did not ubiquitinate GST (Fig 6A), we observed a generation of several high-molecular weight CKAP4 products upon incubation with SneRING (Fig 6B, asterisks). In both instances, SneRING was able to perform autoubiquitination; however, in the presence of CKAP4, this process was inhibited (Fig 6). These experiments confirm that SneRING can, at least *in vitro*, ubiquitinate one of the interaction partners identified in the pull-down assay.

Besides these cellular proteins, a number of *Simkania*-derived proteins were also strongly enriched in the SneRING-pulldowns (S5 Fig and S1 Table). These proteins might be bacterial SneRING targets or co-factors of SneRING. None of the top-enriched Sne proteins appears to be experimentally characterized, but several members of this set have informative sequence relationships. SNE_A02810, SNE_A02800, SNE_A02780, SNE_A18120, and SNE_A05090 are relatives of the *Legionella* major outer membrane protein mOMP, and SNE_08170 contains mOMP-associated POTRA domains. Other enriched proteins are related to components of bacterial secretion systems: A21850 is related to the SecDF secretion protein, SNE_A20110 is related to the T2SS secretin GspD, while SNE_A10070 is related to bacterial signal peptidases. Overall, the bacterial SneRING-associated proteins appear to be located at the bacterial surface.

Taken together, these results show clustering of putative cellular SneRING targets in groups of ER and mitochondrial proteins functioning in quality control, bioenergetics, and morphology. The localization of these targets is in agreement with the possible localization of SneRING that we previously determined (Fig 4). The observation that most of the prominent SneRING interactors are only present in the infected samples suggests that SneRING needs additional bacterial factors for its function, or that the infection-induced remodeling of cellular compartments fosters SneRING interactions.

## Discussion

After invading the host cell, intracellular bacteria are targeted by the innate immune system to induce their clearance and block their propagation. To survive and replicate, intracellular pathogens must modify and/or evade the host defense system. An important part of intracellular defense is the ubiquitin system.

The ubiquitin system has been thought to be exclusive to eukaryotes; however, recent reports about bacterial ubiquitination systems that serve as anti-viral defense show that ubiquitination might be evolutionarily older than presumed [59,60]. In addition, various bacteria specifically counteract their ubiquitination and even hijack the system for their own advantage by secreting effector proteins [61]. Thus, the ubiquitin system of the host cell represents a weapon for both the host cell and the bacteria.

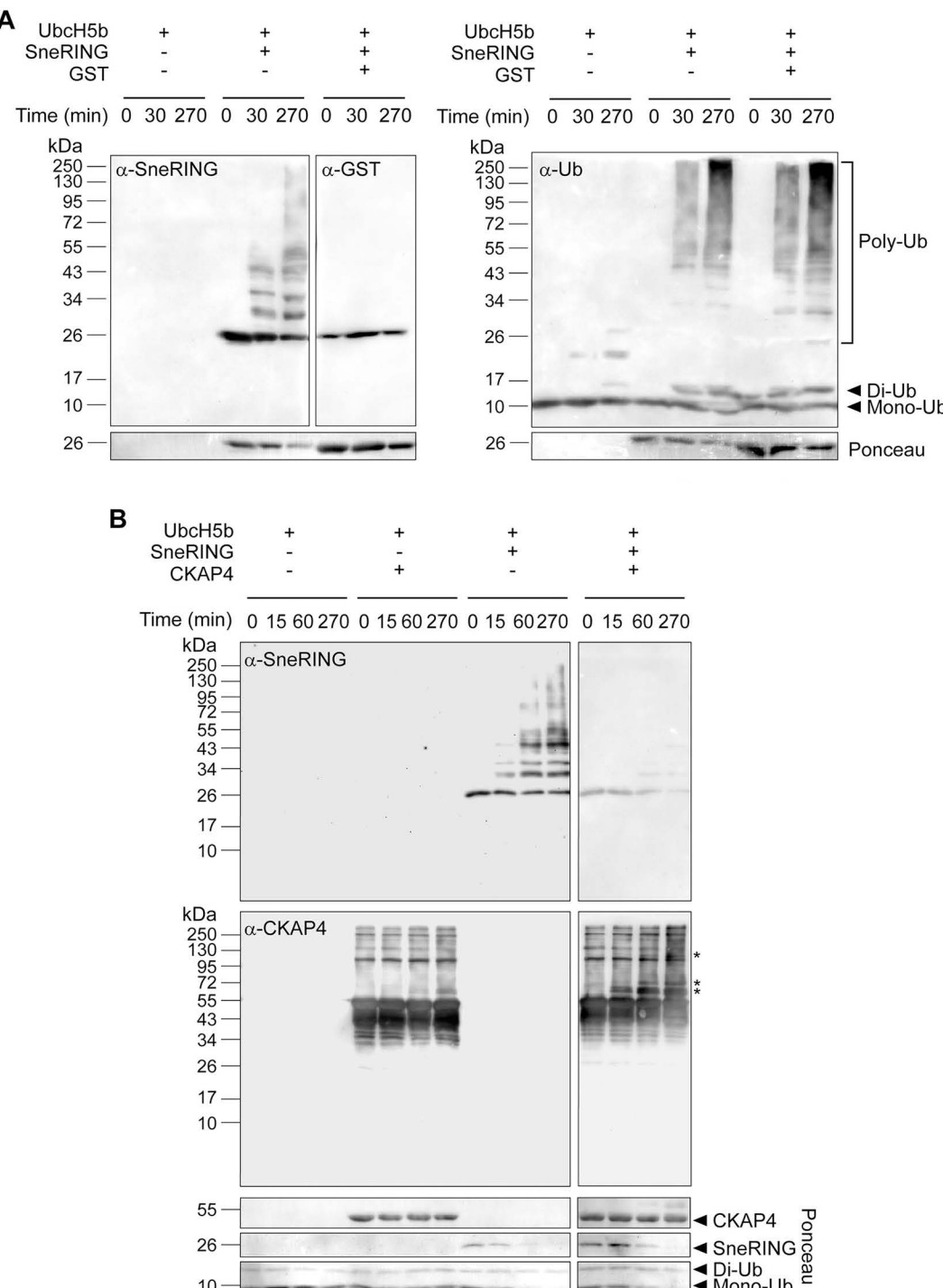

**Fig 6. SneRING ubiquitinates CKAP4 *in vitro*. (A, B)** A reaction containing purified recombinant SneRING, recombinant E1 ubiquitin-activating enzyme, E2 ubiquitin-conjugating enzyme (UbcH5b), ubiquitin, and ATP was mixed with the respective substrate (purified recombinant GST **(A)** or His-CKAP4 **(B)**) for the indicated periods at 37 °C. Reactions without the ligase and/or substrate served as controls. Samples were analyzed by SDS-PAGE and western blot, using antibodies against SneRING, GST, ubiquitin, and CKAP4. Asterisks indicate ubiquitinated CKAP4. Ponceau staining of the western blot membranes is shown for the loading control.

Bacterial DUBs can prevent bacterial ubiquitination, which can be a signal for xenophagy [62–64]. Recent studies have shown that *S. negevensis* possesses a high number of DUBs from many different classes, which is in strong contrast to only one or two DUBs encoded by most other intracellular bacteria [28]. Furthermore, ubiquitin E3 ligases of several types have been described for various bacteria. These differ in their target in the host cell and their intracellular function [65,66]. So far, however, no Sne ubiquitin ligase has been described.

Our bioinformatic analysis revealed a number of genes in Sne that contain a RING domain. In the case of SneRING, the RING as well as the C-terminal domain is shared with related proteins from other *Chlamydia*-like bacteria, with notable differences in the Zn-binding residues of the Sne RING finger. The comparison with two established human RING fingers structurally corresponding best to the SneRING model in the alignment showed that PDB:3LRQ_A (from TRIM37) structurally matches SneRING best; however, it is more similar to the RING finger of the *Chlamydia*-like bacteria than of Sne in terms of Zn binding (Fig 1). In addition to SneRING, Sne possesses several more proteins with a RING domain (S1 Fig), which remain to be characterized.

Most bacterial ubiquitin ligases are of the transthiolating type, including the so-called HECT-like family (bHECT) and novel E3 ligase (NEL) family. Members of the HECT-like family include SopA from *Salmonella* enterica Typhimurium, targeting immune signaling through the inhibition of TRIM56 E3 ligase activity and induction of TRIM56 and TRIM65 degradation [67], and NleL from enterohemorrhagic *Escherichia coli* (EHEC), thought to be involved in pedestal formation and disruption of c-Jun NH2-terminal kinase activation [68,69]. NEL family members were first described in *Shigella flexneri* (IpaH family), but these ubiquitin ligases include related *S.* Typhimurium effectors SspH1, 2, and 3, and SlrP, as well as ubiquitin ligases from *Pseudomonas*, *Sinorhizobium*, and *Ralstonia* species. NEL ubiquitin ligases from *Shigella* and *S.* Typhimurium are connected to the modulation of the inflammatory response. Most of these effectors have been shown to be translocated by the type 3 secretion system (T3SS) [68].

There is limited knowledge about bacterial RING and related U-box ligases. AvrPtoB/HopAB3 from *Pseudomonas syringae*, a plant pathogen, is a RING ligase and inhibitor of host cell death [70,71]. This T3SS effector autoubiquitinates in the presence of UbcH5a but not UbcH7 [70], similar to SneRING (Fig 2). The RING ligase activity of AvrPtoB has been linked to pathogenicity and inhibition of programmed cell death, and the protein has been shown to interact with FLS2, a pathogen-associated molecular pattern (PAMP) sensor, leading to its degradation [70–72]. LubX/LegU2 is a *Legionella* E3 ligase with multiple RING-like U-box domains. It is a substrate of the Dot/Icm type 4 secretion system (T4SS) and translocates into the host cells, possibly mediating polyubiquitination of host Cdc2-like kinase 1 (Clk1) [37]. LubX shows ligase activity in the presence of UbcH5a or UbcH5c E2 enzymes, but is also active in combination with several UBE2D and UBE2E, as well as UBE2W1 enzymes [37,73,74]. Subsequently, however, the LubX interaction with another *Legionella* effector, SidH, has been demonstrated, showing that LubX promotes SidH degradation in the later stages of infection [73]. Other reported U-box ligases include NleG effectors from EHEC, which prefer UBE2D and UBE2E families of host E2 enzymes [75]. Its targets include MED15 subunit of the Mediator complex, Hexokinase-2, and SNAP29 [76], as well as RavN [77] and Golgi-targeted GobX [78], both from *Legionella*. An interesting example from *S.* Typhi, StoD, is also a U-box E3 ubiquitin ligase most active in the presence of UBE2E1, but which can also bind K63- and K48-linked diubiquitin, indicating possible recognition of pre-ubiquitinated targets [79].

These examples illustrate the diversity of the mode of action of bacterial E3 ligases, which can target immune signalling or protect and support the replicative niche. Considering the similarities between Sne and *Legionella* lifestyle, which include intracellular replication and survival within amoeba as a host, it is interesting to note the large number and variety of ubiquitin-modifying enzymes in these bacteria [29,80]. Manipulation of the host ubiquitin system seems to be a common denominator affecting intracellular survival, and SneRING might play an important role in it, targeting host cell proteins, but maybe even bacterial effectors similar to *Legionella* LubX.

As with other intracellular bacteria, the Sne replicative niche SnCV can be a direct target for ubiquitination. This could lead to its complete or partial degradation by autophagy [29]. Twelve identified Sne DUB candidates, of which eight are

active, might play an important role in preventing autophagy by removing ubiquitin from the SnCV surface. The role of SneRING, on the other hand, can be to modulate protein stability or activity through ubiquitination. While some Sne DUBs do not show cleavage specificity, others are only active against certain ubiquitin chain types. Interestingly, the two Sne-encoded DUBs SnJos1 and SnJos2 efficiently cleave K6-, K11-, K48- and K63-linked ubiquitin chains, with another enzyme selectively cleaving K6-linked chains [28]. One possibility is that SneRING generates these specific ubiquitin conjugates in the early stages of infection for them to be removed at later stages by Sne DUBs to support bacterial development and/or release. However, since genetic modification of Sne has so far not been achieved, we are currently unable to confirm to what extent infectivity and life cycle depend on the activity of SneRING and similar Sne ubiquitin-modifying enzymes.

SnCV has a unique tubular morphology and is in close contact with the ER. Its development depends on retrograde vesicular transport [24], and it is presumed that the majority of lipids for the growth of this complex membranous compartment are derived from the ER. Sne is also capable of suppressing ER stress response [22]. On the other hand, mitochondria intertwine with SnCV only in the early stages of infection, whereas at the later time point, we observe fragmentation and loss of mitochondrial mass. Endogenous SneRING is found in the ER/SnCV membrane [24], whereas the overexpressed SneRING localizes to cytosol, ER, and mitochondria (Figs 4 and S4), including its possible association with the inner mitochondrial membrane (Fig 4D). This, together with the identification of putative targets and interacting factors, might offer a clue about the function of this enzyme.

Mitochondrial depolarization leads to the formation of K6-, K11-, K48-, and K63-chain linkages on the surface of these organelles. Parkin, a mitochondrial protein with E3 ubiquitin ligase activity and the potential to form these same linkages, is involved in mitochondrial ubiquitination, which promotes mitophagy [81]. Recently, Parkin has been shown to directly bind to prohibitin 2 (PHB2) through its RING1 domain and promote K11- and K33-linked ubiquitination of PHB2 [82], which enhances the interaction between PHB2 and the central protein in autophagy, LC3 [83]. SneRING generates the same types of ubiquitin chains in an *in vitro* reaction as Parkin (Fig 3), and one of the highly enriched host cell proteins in the SneRING pulldown fractions is PHB2 (Fig 5 and S1 Table). SneRING might lead to PHB2 ubiquitination to induce mitophagy; however, we did not observe mitochondrial fragmentation in non-infected cells expressing SneRING (S4 Fig), nor could we confirm SneRING interaction with PHB by western blot. On the other hand, we could verify the infection-dependent interaction of Mic60/IMMT with SneRING (Fig 5C). A recent report identified Mic60/IMMT as a target of *Listeria monocytogenes* listeriolysin O (LLO), inducing mitochondrial fragmentation [84], so it is tempting to speculate that Mic60 is also manipulated by Sne, which could explain the observed mitochondrial fragmentation at the later stages of infection. In addition, another highly enriched putative SneRING interactor, BCAP31, is a protein present in ER-mitochondria contact sites, reported to regulate mitochondrial respiration by controlling the mitochondrial import of respiratory complex I subunits [56]. Whether these interactions influence mitochondrial morphology and respiration during Sne infection remains to be elucidated.

Erlin2, a highly enriched hit confirmed by co-precipitation and western blot (Fig 5 and S1 Table), also localizes to the ER and is involved in ERAD [57]. This protein is another potential target of modification by SneRING, considering that bacteria can suppress the ER stress response [22], a process counteracted by the ERAD upregulation [85]. Finally, possible SneRING ubiquitination of CKAP4, a protein involved in the attachment of the ER to microtubules [86], might be required during the growth and organization of the SnCV. CKAP4 has recently been reported to also play a role in ER-phagy, a process triggered by ER stressors, such as tunicamycin [87]. Its targeting by SneRING might be one of the ways for bacteria to interfere with ER stress signaling [22]. Though interesting, and although we could show that CKAP4 can be ubiquitinated by SneRING in an *in vitro* reaction, these theories must be confirmed by analyzing whether any of these proteins are ubiquitinated upon SneRING expression or during infection. In addition, the deletion of SneRING from bacteria could give further information about the function and importance of this protein.

To survive and replicate within their host, intracellular pathogens have evolved various strategies and mechanisms to counteract the host cell's immune system. By secreting effector proteins, pathogens manipulate the host cell to provide better conditions for their survival and propagation. Effector proteins also include ubiquitin-modifying enzymes such as

deubiquitinases and ubiquitin E3 ligases. This study describes for the first time a Sne effector protein, SneRING, that functions as ubiquitin E3 RING ligase with a possible role in remodeling host cell organelles to promote the development of SnCV and accommodate its unique replicative niche.

## Materials and methods

### Ethics statement

Primary cells used in this study originate from leukoreduction system cones made available by the Institut für Klinische Transfusionsmedizin und Hämotherapie of the University of Würzburg, Germany, for research purposes and are not subject to approval by the Ethics Committee of the University of Würzburg.

### Sequence analysis

Sequence alignments were generated using the MAFFT package [88]. Generalized profiles were derived from multiple alignments using pftools [43] and searched against the Uniprot database (https://www.uniprot.org) and the NCBI microbial genome reference sequence database (https://www.ncbi.nlm.nih.gov/genome/microbes) using pfsearchV3 [89]. Structure predictions were run using a local installation of Alphafold 2.3 [44]. Structure comparisons were performed using the DALI software [45]. For modeling the missing $Zn^{2+}$ ions into Alphafold structures, a superposition with the best-scoring $Zn^{2+}$-containing DALI hit was used to copy the zinc ions into the model. Subsequently, an energy minimization was performed using YASARA [90] to improve the sidechain positioning of the $Zn^{2+}$ ligands.

### Cloning and mutagenesis

The SneRING coding region for protein purification was obtained by amplification of *S. negevensis* genomic DNA (Leibniz Institute DSMZ, German Collection of Microorganisms and Cell Cultures, DSM No. 27360) using the following primers: forward, 5'-AAGTTCTGTTTCAGGGCCCGGAAAGGGTAAATCCTAATCAAGTTCTA – 3', and reverse, 5'-ATG GTCTAGAAAGCTTTATGAAAAAATTCTACGAGAAATATCTTCC – 3'. Amplification was performed using the Phusion High Fidelity DNA Polymerase Kit (Thermo Fisher Scientific, Massachusetts, USA), and the PCR product was inserted into the pOPIN-K vector [91] using the In-Fusion Snap Assembly Master Mix (Takara Clontech). Point mutations were generated using the QuikChange Lightning kit (Agilent Technologies, California, USA). To introduce the first mutation, the following primers were used: forward, 5' – GCATGGAATTGTCGCAGGTGCTGGGAACCAATTCA – 3', and reverse, 5' – TGAATTG GTTCCCAGCACCTGCGACAATTCCATGC – 3'. The second mutation was added by using the following primers: forward, 5' – AATTGTCGCAGGTGCTGGAGACCAATTCAGGATGGTCC – 3', and reverse, 5' – GGACCATCCTGAATTGGTCTC CAGCACCTGCGACAATT – 3'. SneRING coding region containing a FLAG-tag for transfection experiments was synthesized by Thermo GeneArt Gene synthesis (Thermo Fisher Scientific, Massachusetts, USA) and cloned into the pcDNA3 vector by restriction digestion according to the manufacturer's protocol.

### Protein expression and purification

Wildtype SneRING was expressed from the pOPIN-K vector containing an N-terminal 6His-GST-tag in *E. coli* Rosetta (DE3) pLysS. The bacteria culture was grown in LB medium at 37 °C until reaching an $OD_{600}$ of 0.8. Afterward, the culture was precooled at 18 °C before SneRING expression was induced with 500 μM $ZnCl_2$ and 0.1 mM isopropyl β-d-1-thiogalactopyranoside (IPTG) for 16 h when bacteria were collected by centrifugation at 5,000 x g for 15 min. The pellets were frozen at -80 °C, then thawed on ice and resuspended in binding buffer (300 mM NaCl, 20 mM Tris pH 7.5, 20 mM imidazole, 2 mM β-mercaptoethanol), containing DNase and lysozyme. Cells were sonicated for 10 min with 10 s pulses at 50 W and the lysate was clarified by centrifugation for 1 h at 50,000 x g and 4 °C. The supernatant was applied to a HisTrap FF column (Cytiva, Massachusetts, USA), and affinity purification was performed according to the manufacturer's

protocol. The subsequent incubation of the fractions with 3C protease during dialysis in binding buffer overnight resulted in the 6His-GST tag removal, which was removed together with the His-tagged 3C protease in a second affinity purification run. Size exclusion chromatography (HiLoad 16/600 Superdex 75 pg (Cytiva, Massachusetts, USA)) was performed in a buffer containing 20 mM Tris pH 7.5, 150 mM NaCl, and 2 mM dithiothreitol (DTT). After concentration using VIVASPIN 20 Columns (Sartorius, Göttingen, Germany), the purified protein was frozen in liquid nitrogen and stored at -80 °C. Due to the cloning procedure, the purified protein contained additional Gly and Pro at its amino terminus.

The expression of SneRING S19R_E22R was induced as already described for the wild-type SneRING. After harvesting the cells, the pellet was resuspended in lysis buffer (40 mM Tris pH 8.2, 500 mM NaCl, 10 mM β-mercaptoethanol, 10 mM imidazole, containing 1xcOmplete EDTA-free protease inhibitor cocktail (Roche Holding, Basel, Switzerland), DNase and lysozyme). Cells were lysed by sonication, the cleared lysate was applied to Ni-NTA agarose (Qiagen, Hilden Germany), and affinity purification was performed according to the manufacturer's protocol. Mutated SneRING was eluted using 40 mM Tris pH 8.2, 500 mM NaCl, 10 mM β-Mercaptoethanol, and 250 mM imidazole. The removal of the 6His-GST tag (leaving additional Gly and Pro at its amino terminus) was performed similarly to the wildtype SneRING and using the Ni-NTA agarose (Qiagen, Hilden Germany) in a buffer containing 20 mM Tris pH 7.5 and 300 mM NaCl. After concentration on Amicon Ultra-15, PLGC Ultracel-PL membrane (Sigma/Merck, Darmstadt, Germany), the protein was frozen in liquid nitrogen and stored at -80 °C.

6His-GST, expressed from the pOPIN-K empty vector, and the two Sne DUBs SnVTD and SnOTU used in the study were purified as described before [28]. The plasmids pOPINB-OTUB1* (Addgene plasmid #65441; http://n2t.net/addgene:65441; RRID:Addgene_65441), pOPINB-AMSH* (Addgene plasmid #66712; http://n2t.net/addgene:66712; RRID:Addgene_66712) [92], pOPINK-Cezanne (OTU, aa 53-446), (Addgene plasmid #61581; http://n2t.net/addgene:61581; RRID:Addgene_61581) [93] and pOPINS-USP21 (USP, aa 196-565) (Addgene plasmid #61585; http://n2t.net/addgene:61585; RRID:Addgene_61585) [48] were kind gifts from David Komander (WEHI, Melbourne, Australia). Purification of these DUBs was performed as already described for the SneRING.

Protein concentration was determined by measuring the absorption at 280 nm ($A_{280}$) and using the SneRING extinction coefficient calculated from its sequence.

Recombinant human His-tagged CKAP4 was purchased from Biozol (Eching, Germany) (Cat. No. TGM-TMPK-00936).

### *In vitro* autoubiquitination assay

10 µL of the 10x ligation buffer (800 mM Tris pH 7.5, 200 mM $MgCl_2$, 12 mM DTT) was mixed with 10 µM purified recombinant SneRING, 100 nM recombinant E1 ubiquitin-activating enzyme, and 2.3 µM recombinant E2 ubiquitin-conjugating enzyme, to which 50 µM recombinant ubiquitin and 10 mM ATP were added. ddH$_2$O was added to a final volume of 100 µL. The reactions were incubated at 37 °C for different time points (0 min, 30 min, 60 min, 120 min, 180 min, and 300 min) and stopped by adding Laemmli sample buffer (62.5 mM Tris pH 6.8, 2% SDS, 10% glycerol, 5% β-mercaptoethanol, and 0.002% Bromophenol Blue) and analyzed by sodium dodecyl sulfate-polyacrylamide gel electrophoresis (SDS-PAGE) and western blot.

The purified proteins of the E2 conjugating enzymes UbcH5b, UBE2T, UbcH7, UBE2A, UBE2B, and UBE2W were a kind gift from David Komander (WEHI, Melbourne, Australia).

### *In vitro* substrate ubiquitination assay

MasterMix was prepared, containing 80 mM Tris pH 7.5, 20 mM $MgCl_2$, 1 mM DTT, 0.1 mM mouse E1, 2.3 µM UbcH5b, 5 µM SneRING, 1–5 µM substrate (recombinant His-CKAP4 or recombinant GST), 50 µM ubiquitin, and 10 mM ATP. Reaction was incubated at 37 °C. Per timepoint, reaction was stopped by combining 10 µL MasterMix with 10 µL 2x Laemmli buffer. 10 µL of the stopped sample were analyzed by SDS-PAGE and western blot.

## UbiCRest assay

The recombinant DUBs were first diluted in a buffer containing 150 mM NaCl, 20 mM Tris pH 7.5, and 10 mM DTT. The 5 h *in vitro* autoubiquitination assay reaction was treated for 15 min at RT with 2 mU apyrase to remove the remaining ATP. For the assay, the DUBs were added at a 1.5 µM concentration except OTUB1, for which a 3 µM concentration was used. During the double UbiCRest assay, the DUBs were added at 0.75 µM or 1.5 µM concentrations. The mixture was incubated for 1 h at 37 °C, and the reaction was stopped by adding Laemmli sample buffer.

## Immunoblotting

The lysates for western blot analysis were prepared with Laemmli sample buffer and denatured at 95 °C for 5 min. After SDS-PAGE, proteins were transferred onto a PVDF membrane. The membrane was blocked with 5% milk in Tris-based saline (TBS) and incubated with the primary antibody, followed by incubation with an HRP-coupled secondary antibody (1:3000) in blocking solution. The signal was detected using SuperSignal West Femto Maximum Sensitivity Substrate (Thermo Fisher Scientific, Massachusetts, USA) and a Chemo Cam Imager (Intas). In the case of a K6 affimer, the membrane was incubated overnight at 4 °C in the affimer dilution in blocking solution. The next day, the membrane was washed with TBS and incubated with a 6xHis primary antibody, followed by the HRP-coupled secondary antibody as already described.

## Cell culture

HeLa (ATCC CCL-2.1), HeLa229 KDEL-dsRed, Hela 229 Mito-GFP, and THP-1 (ATCC TIB-202) cells were grown in Roswell Park Memorial Institute (RPMI) 1640 medium (Thermo Fisher Scientific, Massachusetts, USA), Hek293T cells (ATCC CRL-3216) were grown in Dulbecco's modified Eagle's medium (DMEM) medium (Thermo Fisher Scientific, Massachusetts, USA) and U2OS cells (ATCC HTB-9.6) were grown in DMEM/F12 medium (Thermo Fisher Scientific, Massachusetts, USA). All media were supplemented with 10% v/v heat-inactivated (56 °C for 30 min) fetal calf serum (FCS) (Sigma/Merck, Darmstadt, Germany). Growth took place at 37 °C and 5% $CO_2$. To differentiate THP-1 cells into macrophages, 5x $10^5$ cells/well were seeded into 6-well plates and incubated with 20 ng/mL phorbol 12-myristate 13-acetate (PMA) (Sigma/Merck, Darmstadt, Germany) in RPMI1640 medium containing 10% v/v heat-inactivated FCS for 24 h. The medium was exchanged for fresh RPMI1640 medium containing 10% v/v heat-inactivated FCS and the cells were incubated for a further 48 h.

Primary human macrophages were derived from peripheral blood mononuclear cells (PBMCs) isolated from leukoreduction system (LRS) cones, using the SepMate-50 system (#85450, STEMCELL Technologies, Cologne, Germany) and Ficoll-Paque (#GE17-1440-03, Sigma/Merck, Darmstadt, Germany) gradient according to manufacturer's instructions. Red blood cells were removed by hypo-osmotic shock: PBMC pellets were mixed and incubated with 16 mL sterile water for 30 s, then immediately 4 mL 5 x DPBS were added. Monocytes were purified from the PBMC fraction using the EasySep Human CD14 Positive Selection Kit II (# 17858, STEMCELL Technologies, Cologne, Germany) according to the manufacturer's instructions and seeded in 6-well cell culture plates (1.5 x 106 cells/well) in medium supplemented with 50 ng/mL M-CSF (#78057, STEMCELL Technologies, Cologne, Germany). Medium exchanges, with fresh M-CSF supplementation, were done on days 1 and 4. On day 7, "M2"-like polarization, was induced by replacing M-CSF-containing medium with medium containing 100 ng/mL IL-4 (#78045, STEMCELL Technologies, Cologne, Germany) and further incubation for 48 h. All cultivation steps were done in RPMI1640 (Thermo Fisher Scientific, Massachusetts, USA) supplemented with 10% (v/v) heat-inactivated FCS (Sigma/Merck Darmstadt, Germany). All Sne infection experiments were performed on day 9, without cytokine supplementation.

*Acanthamoeba castellanii* (ATCC 30010D) were cultured in axenic medium (56 mM glucose, 2.6 mM KH2PO4, 1.3 mM Na2HPO4 x 7H2O, 0.5% proteose peptone 2, 0.5% thiotone E peptone, 0.5% yeast extract) in cell culture flasks at 30 °C.

## Bacteria and infection

*E. coli* DH5α and Rosetta (DE3) pLysS were grown in an LB medium containing the respective antibiotics for selection (1% bacto-tryptone, 0.5% yeast extract, 1% NaCl, 1% agar).

To prepare *S. negevensis* (ATCC VR-1471), U2OS cells were grown to 60% confluency, and infected at a multiplicity of infection (MOI) 1 in RPMI1640 medium without HEPES containing 5% v/v heat-inactivated FCS (infection medium). The infected cells were incubated at 35 °C and 5% $CO_2$ for 6 h, when the medium was exchanged for fresh infection medium. After 3 days of growth under the same conditions, the cells were detached using a cell scraper and lysed by vortexing the cells with 2–5 mm glass beads (Carl Roth, Karlsruhe, Germany). Intact cells and larger cell debris were removed by low-speed centrifugation at 600 x g for 10 min and the resulting supernatant was centrifuged at 20,000 x g for 30 min to pellet the released bacteria. The pellet was resuspended in 5 mL SPG buffer (250 mM sucrose, 4 mM monopotassium phosphate, 10 mM disodium phosphate, and 5 mM glutamate pH 7.4), followed by centrifugation (20,000 x g, 30 min), with the resulting pellet resuspended in 2.5 mL SPG buffer. The bacteria were homogenized using a syringe with one G20 and one G18 needle, aliquoted, and stored at -80 °C. All the experiments with *S. negevensis* were performed in a laboratory with biosafety level 2, which is registered with the Lower Franconia Government under code 55.1-8791.1.30.

For infection experiments, 8 x $10^4$ or 5 x $10^5$ human cells/well were seeded into 12-well or 6 well plates, respectively, and at 60% confluency infected with *S. negevensis* at MOI 1 in the infection medium, at 35 °C and 5% $CO_2$. 6 h later, the medium was exchanged for fresh infection medium. The infection proceeded for the indicated time points. When the infection took longer than 3 days, the medium was exchanged for fresh infection medium after every 3 days.

*A. castellanii* were seeded into cell culture flasks and infected with *S. negevensis* in an axenic medium. Infected amoebae were incubated at 30 °C for 4 days.

## Transfection

Hek293T cells were transfected using the calcium-phosphate transfection method as described previously [94]. For 6-well plates, 4 µg of the respective plasmid were used; for transfecting cells in 15 cm dishes, we used 50 µg of plasmid DNA. U2OS and HeLa229 cells were transfected using the commercially available transfection reagents jetOptimus (Polyplus, Illkirch-Graffenstaden, France) or PEI MAX (Polysciences, Inc., Warrington, United Kingdom) according to the manufacturer's protocol. For 12-well plates, 1 µg (jetOptimus) or 1.5 µg (PEI MAX) of plasmid per well was used. For 10 cm dishes, initially seeded with 1.2x$10^6$ cells, we used 10 µg of plasmid (PEI MAX). Depending on the experiment, cells were infected with *S. negevensis* at MOI 1, 24 h post-transfection, and the infection was allowed to proceed for 2 days before sample collection.

## Immunofluorescence and Microscopy

U2OS and HeLa cell lines were seeded at a density of 8 x $10^4$ cells/well into 12-well plates on 15 mm glass coverslips (Paul Marienfeld, Lauda-Königshofen, Germany). Cells were transfected and/or infected as indicated in the corresponding experiments. The cells were fixed at respective time points using 4% paraformaldehyde (PFA) in PBS for 1 h at RT. For *S. negevensis* infection experiments, the cells were permeabilized with 0.2% Triton X-100 in PBS for 45 min at RT while shaking, washed with PBS, and blocked in 2% goat serum in PBS for 1 h, followed by staining with a SnGroEL and FLAG primary antibodies in the blocking solution overnight at 4 °C. The following day, samples were washed and incubated with the mix of the respective secondary antibodies (1:100) and DAPI (1:3000) for 1 h in 2% goat serum in PBS at RT. After washing with PBS and dd$H_2$O, coverslips were mounted using Mowiol 4–88 (Carl Roth GmbH + Co. KG, Karlsruhe, Germany).

For immunofluorescence stainings that did not include staining of bacterial proteins, after fixing, cells were simultaneously permeabilized and blocked for 30 min at RT in blocking solution (10% BSA, 0.2% Triton X-100, in PBS), washed with

PBS and stained with the primary antibody diluted in antibody dilution solution (3% BSA in 0.05% Tween in PBS) for 1 h at RT. After washing with PBS and blocking for 10 min in blocking buffer at RT, samples were washed again and incubated with DAPI (1:3000) and the secondary antibodies (1:100) for 1 h at RT. The coverslips were then washed with PBS and ddH$_2$O and mounted using Mowiol 4–88 (Carl Roth GmbH + Co. KG, Karlsruhe, Germany). Microscopy was performed using a LEICA SP5 confocal laser scanning microscope.

### RNA isolation, cDNA synthesis, and RT-qPCR

Infected macrophages were collected on days 2, 4, and 6 p.i. The cells were washed with DPBS at the desired time point, and 100 μL of fresh medium and 500 μL RNAprotect Cell Reagent (Qiagen, Hilden, Germany) were added per well. Detached cells were collected in RNase-free tubes and stored at 4 °C for a maximum of 1 week. After centrifugation at 6200 x g for 5 min at 4 °C, 600 μL RLT Buffer supplemented with β-mercaptoethanol was added to each pelleted sample, and RNA was purified using the RNeasy Mini kit (Qiagen, Hilden, Germany) according to the manufacturer's instructions. For other human cell lines and amoeba, which were collected on day 2 and 4 p.i., or just on day 4 p.i., respectively, and washed with DPBS at the desired time point, 600 μL RLT Buffer supplemented with β-mercaptoethanol was added to each sample and the RNA was purified using the RNeasy Mini (Qiagen, Hilden, Germany) according to manufacturer's instructions. DNase treatment was performed using the TURBO DNA-free Kit (Invitrogen, California, USA). 100 ng of pure RNA was used for cDNA generation, using the Revert Aid First Strand cDNA Synthesis Kit (Thermo Fisher Scientific, Massachusetts, USA). RT-qPCR was performed on a StepOnePlus instrument (Life Technologies, California, USA), using the GreenMasterMix (2X) High ROX (Gennaxon, Ulm, Germany). Data analysis was performed using the StepOne Software (v2.3) and Microsoft Excel, with a modified ($2^{-\Delta\Delta Ct}$) method [95]. ΔCt values were calculated relative to the host reference gene *YWHAZ* (human) or the 5S RNA (for amoeba). Fold changes ($2^{-\Delta\Delta Ct}$) of gene expression for each tested Sne gene, were calculated relative to the non-infected controls. The following primers were used: forward, 5' – ATACTGCGCAGAAAG CAC – 3', and reverse, 5' – ACCCCAGTACTAACACCG – 3' (5S RNA reference gene for amoeba) [28]; forward, 5' – GCAGAGTCGAAGTTTCAAGGG – 3', and reverse, 5' – TCGCTTGCCTGTTGAGGATT – 3' (SneRING); forward, 5' – TTCCATCACCGGCAACATCA – 3', and reverse, 5' – GCAAAAGAGATCGCGCTGAA – 3' (GroEL C (SNE_A22590)) [28]; forward, 5' – CACCTGATCCCATCCCGAAC – 3', and reverse, 5' – GCGACCTACTCTCCCGTACT – 3' (Sne 5S RNA) [28]; forward, 5' – ACTTTTGGTACATTGTGGCTTCAA – 3', and reverse, 5' – CCGCCAGGACAAAC CAGTAT – 3' (*YWHAZ*) [96].

### Mitochondrial isolation and fractionation

Mitochondria were isolated from cells transfected for 48 h as described previously [97]. Cellular fractionation and determination of submitochondrial localization by swelling and PK treatment were performed as already described [98,99]. Carbonate extraction for determining protein membrane extractability was performed as previously described [100]. In brief, isolated mitochondria from transfected cells were either sonicated in 10 mM Tris pH 7.6 and 500 mM NaCl or treated with 100 mM Na$_2$CO$_3$ at pH 10.8 or 11.5 for 20 min on ice. After centrifugation at 100.000 g, pellets were resuspended in Laemmli sample buffer, while supernatant and non-centrifuged control samples were precipitated using trichloroacetic acid (TCA).

### Antibodies, chemicals, and controls

K6-, K48-, and K63-linked ubiquitin chains used as a control in S2 Fig were components of the Di-ubiquitin Explorer Panel purchased from UbiQ (Amsterdam, Netherlands, Cat. No. UbiQ-L01). M1-linked ubiquitin chains (kindly provided by S. Meyer and A. Buchberger) were produced by incubating the following reaction mixture at 37 °C for 2 h: 6 μg bovine ubiquitin, 200 ng yeast Uba1, 600 ng human UbcH5c, 4 μg HOIP RBR-LDD [101] in a buffer containing 50 μL 20 mM Tris/HCl pH 7.5, 5 mM MgCl$_2$, 1 mM DTT, and 4 mM ATP.

Primary antibodies against ubiquitin (Cat. No. 05-944), FLAG-tag (Cat. No. F7425; Cat. No. F3165), CPOX (Cat. No. HPA015736), K63-linked ubiquitin chains (Cat. No. 05-1308), K48-linked ubiquitin chains (Cat. No. 05-1307), M1-linked ubiquitin chains (Cat. No. MABS451, kindly provided by A. Buchberger), and the anti-ubiquitin Lys6 specific Affimer (Cat. No. MABS1918, kindly provided by M. Eilers) were purchased from Sigma/Merck (Darmstadt, Germany). Primary antibodies against Tubulin (Cat. No. ab18251) and Mic60 (Cat. No. ab48139) were purchased from Abcam (Cambridge, United Kingdom). Primary antibodies against Calnexin (Cat. No. 2433 and Cat. No. sc-23954) were purchased from Cell Signaling (Massachusetts, USA) and Santa Cruz Biotechnology (Texas, USA), respectively. The Erlin2 antibody (Cat. No. 2959) was purchased from Cell Signaling (Massachusetts, USA). The Prohibitin antibody (Cat. No. sc-377037) and the antibody against GAPDH (Cat. No. sc-47724) were purchased from Santa Cruz Biotechnology (Texas, USA). The anti-Hsp60 antibody (Cat. No. ADI-SPA-806) was purchased from Enzo Life Sciences (New York, USA). The primary antibody against Tim44 (Cat. No. 612582) was purchased from BD Transduction Laboratories (California, USA). The antibody against 6xHis (Cat. No. GTX18184) was purchased from Biozol (Eching, Germany). The antibody against GST (Cat. No. MA4-004) was purchased from Invitrogen (Thermo Fisher Scientific, Massachusetts, USA). The antibody against K11-linked ubiquitin chains (Cat. No. NBP3-05681, kindly provided by A. Buchberger,) and the CoxVa antibody (Cat. No. NBPI-32550) were purchased from Novus Biologicals (Littleton, USA). The CKAP4 antibody (Cat. No. 16686-1-AP) was purchased from Proteintech (Rosemont, USA). The antibody against Sam50 was custom-generated against the full-length protein (Gramsch laboratories, Schwabhausen, Germany). The antibody against rat Tom70 (crossreactive with human Tom70) was a gift from K. Mihara. The antibody against SnGroEL was produced as described in a previous study [22]. The SneRING antibody was custom-made against the 189 carboxy-terminal amino acids of the protein (Davids Biotechnologie GmbH, Regensburg, Germany). Secondary antibodies anti-rabbit-Alexa555, Cat. No. A-21428, anti-mouse-Alexa488 Cat. No. A-11001, and anti-mouse-Alexa555, Cat. No. A-21422 were purchased from Thermo Fisher Scientific (Massachusetts, USA). Anti-rabbit-Atto647N secondary antibody, Cat No. 40839, was purchased from Sigma/Merck (Darmstadt, Germany). The HRP-coupled secondary antibodies for western blot were purchased from Biozol (Eching, Germany).

## Immunoprecipitation

For immunoprecipitation (IP) assays, U2OS cells were seeded at a density of 1.2 x 10$^6$ cells/well into 10 cm dishes. The next day, the cells were transiently transfected with 10 µg of plasmid carrying a FLAG-tagged SneRING gene and 24 h later infected with Sne at an MOI 1. Non-transfected and non-infected cells served as controls. On day 2 p.i., the cells were washed once with PBS, harvested in lysis buffer (20 mM Tris pH 7.5, 150 mM NaCl, 0.1% NP-40, proteinase inhibitor (Roche: cOmplete, EDTA-free Protease Inhibitor Cocktail), 1 mM PMSF, 5 mM Iodoacetamide)), incubated for 30 min at 4 °C and sonicated for 15 min using an ultrasonic water bath, followed by a 30 min incubation on a rotary shaker at 4 °C. After removing the cell debris by centrifugation (15 min, 14000 x g, 4 °C), the supernatant was added to equilibrated Anti-FLAG M2 magnetic beads (Sigma/Merck, Darmstadt, Germany) and incubated on a rotary shaker overnight at 4 °C. The beads were washed three times for 5 min at 4 °C with washing buffer (20 mM Tris pH 7.5, 150 mM NaCl, 0.1% NP-40) and once with a buffer containing 20 mM Tris pH 7.5, 150 mM NaCl. For mass spectrometry analysis, bound proteins were eluted with 200 µg/mL FLAG peptide (Sigma/Merck, Darmstadt, Germany) in elution buffer (6 M urea, 2 M thiourea).

## Mass spectrometry analysis

For in-solution digest, the samples were reduced by adding 1 µL of 1 M DTT for 1 h at RT, alkylated with 1 µL of 400 mM chloroacetamide (CAA) for 45 min at RT in darkness, digested with 1 µL of the Lysyl endopeptidase (Lys-C, 0.5 µg/µL) for 3 h at RT, treated with 270 µL of 50 mM triethylammonium bicarbonate buffer and finally digested with 1 µL of trypsin enzyme (1 µg/µL) overnight at RT. The next day, samples were acidified by adding 1 µL of 100% formic acid. After in-solution digestion, peptides were bound to styrene-divinylbenzene reverse-phase stage tips (SDB-RP). One stage tip was used per sample. The stage tips were equilibrated by adding 20 µL of 100% methanol and centrifuging for 1 min at 700 x

g. Afterward, 20 µL of buffer B (0.1% formic acid in 80% acetonitrile) was added, and the stage tips were again centrifuged for 1 min at 700 x g. In the next step, 20 µL of buffer A (0.1% formic acid in water) was added to each stage tip, the samples were centrifuged for 1 min at 700 x g, and an additional 20 µL of buffer A was added. Acidified samples were spun down for 5 min at 17,000 x g, the supernatant was loaded onto the SDB discs of the equilibrated stage tips, and they were again centrifuged for 5 min at 700 x g. The stage tips were washed with buffers A and B and completely dried with a syringe.

After binding peptides to SDB-RP Stage tips, they were analyzed by liquid chromatography-mass spectrometry/mass spectrometry (LC-MS/MS). LC-MS/MS analysis was performed by the Proteomics Facility of the CECAD, University of Cologne, as described previously [102]. Briefly, for LC-MS/MS analysis, an EASY-nLC 1000 chromatograph (Thermo Scientific) was coupled to the quadrupole-based Q Exactive Plus (Thermo Scientific) instrument by a nano-spray ionization source. Peptides were separated on a 50 cm in-house-packed column using a two-solvent buffer system: buffer A (0.1% formic acid) and buffer B (0.1% formic acid in acetonitrile). The amount of buffer B was increased from 7% to 23% within 40 min, followed by an increase to 45% in 5 min, and a washing and re-equilibration step before the next sample injection. The mass spectrometer operated in a Top 10 data-dependent mode, using the following settings: MS1: 70,000 (at 200 m/z) resolution, 3e6 AGC target, 20 ms maximum injection time, 300–1750 scan range; MS2: 35,000 (at 200 m/z) resolution, 5e5 AGC target, 120 ms maximum injection time, 1.8 Th isolation window, 25 normalized collision energy. Data analysis was performed by MaxQuant software, V 1.5.4.742, using the Andromeda search engine against the human proteome reference data (including splice variants) from UniProt. The default mass tolerance and modification settings were used. Re-quantify, label-free quantification, and match between runs were enabled. 'Oxidation on M', 'Phosphorylation on S,T,Y', and 'GlyGly on K' were allowed as variable modifications. Intensities were averaged over biological triplicates, and the log2 of the intensity ratio 'sample average/control average' was used for enrichment quantification. To account for missing values, pseudo counts corresponding to the minimal observed intensity were added to the sample and control averages. Statistical analysis was performed using a two-tailed Student's T-test (n = 3).

## Supporting information

**S1 Fig. SNE_A08700 and SNE_A19470 are two other RING-like E3 ligase candidates from *S. negevensis*.** (A) The Alphafold model of SNE_A08700 shows an N-terminal RING-like domain (blue) with a large insertion after the third Zn-coordinating residue (grey). The $Zn_1$ and $Zn_2$ ions coordinated by the RING-like domain are shown in red and magenta, respectively. (B) Multiple alignment of the RING domains of SNE_A08700, SNE_A08690, and some bacterial relatives (Chlr-KDK64: Chlamydiia bacterium, Uniprot: A0A960RSF6; Ver-JSS10: Verrucomicrobiota bacterium, Uniprot: A0A9E0XGC6; Neptunochl: Candidatus *Neptunochlamydia* sp., RefSeq: WP_316358199) and the sequence of the best DALI hit (pdb:5DKA). Residues invariant or conserved in at least 50% of the sequences are shown on black and grey background, respectively. Residues involved in the coordination of $Zn_1$ and $Zn_2$ are highlighted in red and magenta, respectively. (C) Alphafold model of the RING-like domain of SNE_A19470 shows a divergent RING fold (blue) where only the ligands of the $Zn_2$ ion (magenta) are conserved. (D) Multiple alignment of the RING-like region of SNE_A19470 with the two best DALI hits (pdb:8A38 and pdb:6YXE). Coloring as in B.
(TIF)

**S2 Fig. Immunoblot detection of ubiquitin-linked chains generated by SneRING.** (A-F) *In vitro* autoubiquitination assay, as in Fig 2A, was performed for 5 h at 37 °C. The reaction was analyzed by SDS-PAGE and western blot, using primary antibodies against (A) SneRING, (B) K63-, (C) K48-, (D) K11-, and (F) M1-linked ubiquitin chains. For the detection of K6-linked ubiquitin chains (E), a His-tagged K6 affimer was used, subsequently detected by His-tag antibody. Positive controls were purchased (K63-, K48-, and K6-linked chains in B, C and E, respectively) or synthesized *in vitro* (M1-linked ubiquitin chains, F).
(TIF)

**S3 Fig. SneRING is expressed during infection in different host cells.** (A, B) HeLa229 or U2OS cells were infected with Sne at an MOI 1, and RNA was isolated on days two and four p.i. (C) THP-1 cells were differentiated into macrophage-like cells using PMA, infected with Sne, and RNA was isolated on days two and four p.i. (D) Primary human "M2"-like macrophages were derived from peripheral blood monocytes (M-CSF/IL-4). After infection with Sne, RNA was isolated on days two, four, and six p.i. (E) *A. castellanii* was infected with Sne, and four days p.i., RNA was isolated. A modified $2^{-\Delta\Delta Ct}$ method was used to quantify the expression of SneRING and SnGroEL for each time point. ΔCt values were calculated relative to the human reference gene YWHAZ (A, B, C, and D) or 5S RNA of *A. castellanii* (E). $\log_2$ fold change ($\log_2 FC/2^{-\Delta\Delta Ct}$) for each tested Sne gene vs the non-infected control sample was calculated. Expression levels relative to Sne 5S RNA are shown as mean ± SD from independent biological replicates (n = 3 or n = 5). Raw data of the graphs can be found in S2 Table.
(TIF)

**S4 Fig. SneRING distribution in cells after expression.** (A) FLAG-tagged version of the SneRING was transfected into HeLa229 cells expressing mitochondria-targeted GFP (Mito-GFP, green channel). 24 h after transfection, the cells were either left uninfected (-Sne) or were infected with Sne at an MOI of 1 (+Sne). On day 3 p.i., the cells were fixed and stained with DAPI (blue channel), and primary antibodies against the FLAG-tag (red channel) and SnGroEL (magenta channel), followed by a fluorophore-coupled secondary antibody. (B) HeLa229 cells stably expressing ER-targeted KDEL-dsRED (red channel) were transfected with a FLAG-tagged version of the SneRING. 24 h post-transfection, one set of samples was infected with Sne for 3 days (+Sne), when they were fixed together with control, non-infected cells (-Sne), and stained using DAPI (blue channel), and antibodies against the FLAG-tag (green channel) and SnGroEL (magenta channel), followed by staining with fluorophore-coupled secondary antibodies. All images were taken using laser confocal scanning microscopy. The scale bar is 10 µm.
(TIF)

**S5 Fig. Potential *S. negevensis* proteins that interact with SneRING.** The graph shows identified bacterial proteins from samples described in Fig 5A, with significance (-$\log_{10}$p-value calculated using two-tailed Student's T-test, n = 3) plotted against the $\log_2$ fold change ($\log_2 FC$) of transfected/infected U2OS cells (U2OS+Sne + SneRING) relative to non-transfected/infected controls (U2OS+Sne). Enriched proteins are presented in red, reduced proteins are labeled in blue, and unchanged proteins are grey.
(TIF)

**S1 Table. List of significantly enriched host cell and bacterial potential interactors of SneRING.** Significantly enriched proteins in pull-down samples described in Figs 5 and S5 with $\log_2$ fold change > 2 are listed in descending order. A darker color shade represents higher values. SneRING (SNE_A12920) is highlighted in yellow.
(XLSX)

**S2 Table. Data of the RT-qPCR graphs presented in S3 Fig.**
(XLSX)

## Acknowledgments

We thank Prof. Dr. David Komander from the University of Melbourne for providing a panel of deubiquitinating and E2 ubiquitin-conjugating enzymes. We thank Prof. Dr. Alexander Buchberger and Prof. Dr. Martin Eilers from the University of Würzburg and Prof. Dr. Katsuyoshi Mihara from Kyushu University for the antibodies. We also thank Susanne Meyer and Alexander Buchberger from the University of Würzburg for the production of M1-linked ubiquitin chains.

## Author contributions

**Conceptualization:** Vera Kozjak-Pavlovic.

**Formal analysis:** Eva-Maria Hörner, Adriana Moldovan, Kay Hofmann, Vera Kozjak-Pavlovic.

**Funding acquisition:** Vera Kozjak-Pavlovic.

**Investigation:** Eva-Maria Hörner, Vanessa Lachmayer, Thomas Hermanns.

**Methodology:** Eva-Maria Hörner, Vanessa Lachmayer, Thomas Hermanns, Adriana Moldovan, Kay Hofmann, Vera Kozjak-Pavlovic.

**Resources:** Thomas Hermanns, Kay Hofmann, Vera Kozjak-Pavlovic.

**Supervision:** Vera Kozjak-Pavlovic.

**Validation:** Eva-Maria Hörner, Vera Kozjak-Pavlovic.

**Visualization:** Eva-Maria Hörner, Adriana Moldovan, Kay Hofmann, Vera Kozjak-Pavlovic.

**Writing – original draft:** Eva-Maria Hörner, Adriana Moldovan, Kay Hofmann, Vera Kozjak-Pavlovic.

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
