## [Decision Letter · Decision Letter 0]

27 Feb 2025

Identification and characterization of a ubiquitin E3 RING ligase of the Chlamydia-like bacterium Simkania negevensis

PLOS Pathogens

Dear Dr. Kozjak-Pavlovic,

Thank you for submitting your manuscript to PLOS Pathogens. After careful consideration, we feel that it has merit but does not fully meet PLOS Pathogens's publication criteria as it currently stands. Therefore, we invite you to submit a revised version of the manuscript that addresses the points raised during the review process.

Please submit your revised manuscript within 60 days Apr 28 2025 11:59PM. If you will need more time than this to complete your revisions, please reply to this message or contact the journal office at plospathogens@plos.org. Please include the following items when submitting your revised manuscript:

We look forward to receiving your revised manuscript.

Kind regards,

David Skurnik, M.D., Ph.D.

Section Editor

PLOS Pathogens

Editor-in-Chief

PLOS Pathogens

orcid.org/0000-0003-2946-9497

Editor-in-Chief

PLOS Pathogens

orcid.org/0000-0002-7699-2064

**Additional Editor Comments:**

All initial comments from the reviewers will have to be addressed in your revised manuscript.

**Journal Requirements:**

1) Please ensure that the Title in your manuscript file and the Title provided in your online submission form are the same.

https://journals.plos.org/plospathogens/s/submission-guidelines#loc-parts-of-a-submission

- ® on pages: 19, 20, 21, 23, 24, 26, and 27.

- TM on pages: 16, 17, 19, and 20.

5) We have noticed that you have uploaded Supporting Information files, but you have not included a list of legends. Please add a full list of legends for your Supporting Information files after the references list.

6) Please ensure that the funders and grant numbers match between the Financial Disclosure field and the Funding Information tab in your submission form. Note that the funders must be provided in the same order in both places as well. State what role the funders took in the study. If the funders had no role in your study, please state: "The funders had no role in study design, data collection and analysis, decision to publish, or preparation of the manuscript.".

**Reviewers' Comments:**

Reviewer's Responses to Questions

**Part I - Summary**

Reviewer #1: The authors did not even attempt to address the critical points raised by the reviewers.

Reviewer #2: N/A

**Part II – Major Issues: Key Experiments Required for Acceptance**

Reviewer #1: For an E3 ligase, whether its binding partner identified by pulldown or whatever method is physiologically meaningful is whether the target is ubiquitinated and how the ubiquitination benefits the pathogen. If none of these SneRING-binding proteins cannot be ubiquitinated by this E3, they are not relevant to its function as an E3 Ub ligase.

Reviewer #2: I am happy that the authors have made sufficient effort to address my comments. No further revisions are requested.

**Part III – Minor Issues: Editorial and Data Presentation Modifications**

Reviewer #1: (No Response)

Reviewer #2: N/A

PLOS authors have the option to publish the peer review history of their article (what does this mean? ). If published, this will include your full peer review and any attached files.

**Do you want your identity to be public for this peer review?** For information about this choice, including consent withdrawal, please see our Privacy Policy .

Reviewer #1: No

Reviewer #2: No

**Figure resubmission:**

**Reproducibility:**



---

## [Decision Letter · Decision Letter 1]

27 May 2025

Identification and characterization of a ubiquitin E3 RING ligase of the Chlamydia-like bacterium Simkania negevensis

PLOS Pathogens

Dear Dr. Kozjak-Pavlovic,

Thank you for submitting your manuscript to PLOS Pathogens. After careful consideration, we feel that it has merit but does not fully meet PLOS Pathogens's publication criteria as it currently stands. Therefore, we invite you to submit a revised version of the manuscript that addresses the points raised during the review process.

Please submit your revised manuscript within 60 days Jul 26 2025 11:59PM. If you will need more time than this to complete your revisions, please reply to this message or contact the journal office at plospathogens@plos.org. Please include the following items when submitting your revised manuscript:

We look forward to receiving your revised manuscript.

Kind regards,

David Skurnik, M.D., Ph.D.

Section Editor

PLOS Pathogens

Editor-in-Chief

PLOS Pathogens

orcid.org/0000-0003-2946-9497

Editor-in-Chief

PLOS Pathogens

orcid.org/0000-0002-7699-2064

**Additional Editor Comments :**

Performing the experiments suggested by Reviewer 1, or similar experiments, will be necessary for a revised version to be considered for publication.

**Reviewers' Comments:**

Reviewer's Responses to Questions

**Part I - Summary**

Reviewer #1: The authors still did not attempt to address a critically important issue with experiments.

Reviewer #2: I am happy that the authors have made sufficient effort to address my comments. No further revisions are requested.

**Part II – Major Issues: Key Experiments Required for Acceptance**

Reviewer #1: This reviewer fully understands the fact that genetic manipulation of S. negevensis currently is not possible, but this is NOT the question in focus. The question of relevance here is whether any of the interacting proteins identified by the authors is directly ubiquitinated by SneRING. The key experiment is to use biochemical reactions to show that one or more of these proteins are ubiquitinated by SneRING. If recombinant proteins from E. coli cannot be detectably modified by the bacterial E3 in such assays, the authors may obtain the candidate proteins from mammalian cells by transfection and then immunoprecipitation. If these proteins cannot be detectably ubiquitinated by SneRING, the identification of these proteins as the binding targets is not relevant to its activity as an E3.

Reviewer #2: Please above

**Part III – Minor Issues: Editorial and Data Presentation Modifications**

Reviewer #1: (No Response)

Reviewer #2: Please above

PLOS authors have the option to publish the peer review history of their article (what does this mean? ). If published, this will include your full peer review and any attached files.

**Do you want your identity to be public for this peer review?** For information about this choice, including consent withdrawal, please see our Privacy Policy .

Reviewer #1: No

Reviewer #2: No

**Figure resubmission:**

**Reproducibility:**



---

## [Decision Letter · Decision Letter 2]

13 Oct 2025

Dear Dr. Kozjak-Pavlovic,

We are pleased to inform you that your manuscript 'Identification and characterization of a ubiquitin E3 RING ligase of the Chlamydia-like bacterium Simkania negevensis' has been provisionally accepted for publication in PLOS Pathogens.

Best regards,

David Skurnik

Section Editor

PLOS Pathogens

Sumita Bhaduri-McIntosh

Editor-in-Chief

PLOS Pathogens

orcid.org/0000-0003-2946-9497

Michael Malim

Editor-in-Chief

PLOS Pathogens

orcid.org/0000-0002-7699-2064

Reviewer Comments (if any, and for reference):

Reviewer's Responses to Questions

**Part I - Summary**

Reviewer #2: The authors have sufficiently answered all questions and concerns. No further revisions are requested.

**Part II – Major Issues: Key Experiments Required for Acceptance**

Reviewer #2: (No Response)

**Part III – Minor Issues: Editorial and Data Presentation Modifications**

Reviewer #2: (No Response)

PLOS authors have the option to publish the peer review history of their article (what does this mean? ). If published, this will include your full peer review and any attached files.

**Do you want your identity to be public for this peer review?** For information about this choice, including consent withdrawal, please see our Privacy Policy .

Reviewer #2: No

---

## [Editor Report · Acceptance letter]

Dear Dr. Kozjak-Pavlovic,

We are delighted to inform you that your manuscript, "Identification and characterization of a ubiquitin E3 RING ligase of the Chlamydia-like bacterium Simkania negevensis," has been formally accepted for publication in PLOS Pathogens.

Best regards,

Sumita Bhaduri-McIntosh

Editor-in-Chief

PLOS Pathogens

orcid.org/0000-0003-2946-9497

Michael Malim

Editor-in-Chief

PLOS Pathogens

orcid.org/0000-0002-7699-2064